# TRPV3-ANO1 interaction positively regulates wound healing in keratinocytes

Yu Yamanoi[1,2,3], Jing Lei[1,2], Yasunori Takayama[4], Shigekuni Hosogi[5], Yoshinori Marunaka[6,7] & Makoto Tominaga [1,2✉]

Transient receptor potential vanilloid 3 (TRPV3) belongs to the TRP ion channel super family and functions as a nonselective cation channel that is highly permeable to calcium. This channel is strongly expressed in skin keratinocytes and is involved in warmth sensation, itch, wound healing and secretion of several cytokines. Previous studies showed that anoctamin1 (ANO1), a calcium-activated chloride channel, was activated by calcium influx through TRPV1, TRPV4 or TRPA1 and that these channel interactions were important for TRP channel-mediated physiological functions. We found that ANO1 was expressed by normal human epidermal keratinocytes (NHEKs). We observed that ANO1 mediated currents upon TRPV3 activation of NHEKs and mouse skin keratinocytes. Using an in vitro wound-healing assay, we observed that either a TRPV3 blocker, an ANO1 blocker or low chloride medium inhibited cell migration and proliferation through p38 phosphorylation, leading to cell cycle arrest. These results indicated that chloride influx through ANO1 activity enhanced wound healing by keratinocytes.

[1] Thermal Biology Group, Exploratory Research Center on Life and Living Systems (ExCELLS), 5-1 Higashiyama, Myodaiji, Okazaki, Aichi 444-8787, Japan. [2] Division of Cell Signaling, National Institute for Physiological Sciences, 5-1 Higashiyama, Myodaiji, Okazaki, Aichi 444-8787, Japan. [3] Research Laboratory, Ikedamohando Co., Ltd., 16 Jinden, Kamiichi, Nakaniikawa, Toyama 930-0394, Japan. [4] Department of Physiology, Showa University School of Medicine, Tokyo 142-8555, Japan. [5] Department of Clinical and Translational Physiology, Kyoto Pharmaceutical University, 5 Nakauchi-cho, Misasagi, Yamashina-ku, Kyoto 607-8414, Japan. [6] Research Organization of Science and Technology, Ritsumeikan University, Kusatsu 525-8577, Japan. [7] Medical Research Institute, Kyoto Industrial Health Association, Kyoto 604-8472, Japan. ✉email: tominaga@nips.ac.jp

Transient receptor potential (TRP) channels are comprised of six related protein families in mammals. These families include TRPA (ankyrin), TRPC (canonical), TRPM (melastatin), TRPML (mucolipin), TRPP (polycystin), and TRPV (vanilloid). Most are nonselective cation channels and highly permeable to calcium[1]. In choroid plexus epithelial cells, calcium influx through TRPV4 activates a calcium-activated chloride channel (CaCC) termed anoctamin1 (ANO1 or TMEM16A), which is thought to lead to the secretion of cerebrospinal fluid[2]. Other epithelial cells were also reported to have both TRPV4 and ANO1 channels. For instance, these ion channels are co-expressed by both salivary and lacrimal gland acinar cells, and saliva and tear secretions could be accelerated by functional interaction between TRPV4 and ANO1[3]. Thus, TRP channel/ANO1 complexes have distinct functions even though each TRP channel works independently. As a result, one cannot investigate channel functions without studying their interactions when a TRP channel and ANO1 are co-expressed by the same cells.

TRPV3 is a warmth-sensitive TRP channel, and calcium permeability is approximately ten times higher than that of sodium[4]. Although TRPV3 is reportedly expressed throughout the body, its physiological significance is not well understood. Previous studies showed that TRPV3 contributed to itch, warmth sensation, and wound healing in keratinocytes[5–9]. Moreover, TRPV3 activation enhanced by cell signaling downstream of the epidermal growth factor receptor accelerates warmth-dependent wound healing in oral epidermal cells[5]. In skin keratinocytes, a previous study showed that TRPV3 contributes to cell proliferation through EGFR-dependent signaling pathways[5–9].

Wound healing of the skin depends on cell migration and cell proliferation. Although some growth factors and interleukins are involved in wound healing[9,10], ion channels on plasma membranes of keratinocytes could also be important. For example, *Ano1* is reportedly a carcinoma-related gene[11,12], and a recent study revealed that ANO1 was involved in the proliferation of prostate epithelial cells in benign prostatic hyperplasia[13] as well as in HaCaT cells, a special cell line of keratinocytes[14]. In addition, ANO1 inhibition reduced cell migration in some cancer cells[12,15,16]. However, the physiological function of ANO1 in normal skin keratinocytes is not clear even though many epithelial cells express it[17]. Here, we show that ANO1 is expressed in normal human keratinocytes and that these channels are involved in wound healing. This study shows the significance of ANO1 in cell migration and proliferation in normal keratinocytes.

## Results

**Expression of TRP channels and ANOs in normal human epidermal keratinocytes**. The levels of endogenous gene expression of TRP channels and ANOs in NHEKs are unclear. To clarify the expression patterns, we performed RT-PCR analysis of cultured NHEKs (Fig. 1a and Suppl. Fig. 11). Previous reports suggested that TRPV1, TRPV3, TRPV4, TRPV6 proteins were expressed in keratinocytes[18]. Here, we observed mRNAs of *TRPV1, TRPV2, TRPV3, TRPV4, TRPV6, ANO1, ANO4, ANO9,* and *ANO10* (Fig. 1a and Suppl. Fig. 11). Although ANO2 also functions as a CaCC, a discrete band with the predicted molecular weight was not detected in NHEKs. In contrast, ANO1 protein expression was observed by Western blotting (Fig. 1b). In addition, using a high intracellular calcium concentration (500 nM free calcium) with NHEK cells, the currents displayed slow activation kinetics during step-pulses and slow deactivation kinetics upon returning to the holding potential, and outward rectification, characteristic properties of ANO1[19] (Suppl. Fig. 1). These results suggested significant expression of ANO1 in NHEKs. Additionally, we performed calcium-imaging experiments using

Fura-2 to investigate the functional expression of TRP channels. Whereas camphor (10 mM, a TRPV3 agonist) and GSK1016790A (300 nM, a TRPV4 agonist) obviously induced intracellular calcium increases in all cells, capsaicin (300 nM to 3 μM, a TRPV1 agonist), menthol (100 μM, a TRPM8 agonist), allyl isothiocyanate (AITC, 100 μM or 1 mM, a TRPA1 agonist), probenecid (100 μM, a TRPV2 agonist) or 1-oleoyl-acetyl-*sn*-glycerol (OAG, 90 μM, a TRPC6 agonist) did not produce clear responses (Fig. 1c and Suppl. Fig. 2).

We also performed calcium-imaging experiments using a calcium-free extracellular medium because intracellular calcium concentrations are supposed to be reduced upon removal of extracellular calcium in cells expressing TRPV6, which can be constitutively active. However, the intracellular calcium concentrations were not different in the presence or absence of extracellular calcium (Fig. 1d). In addition, a typical TRPV6-mediated current with inward rectification was not observed in NHEKs (Fig.1e). Based on these findings, we concluded that TPRV6 is not functionally expressed in NHEKs. These results indicated that in NHEKs, the most active TRP channels were TRPV3 and TRPV4. However, the expression of other TRP channels was suggested in RT-PCR experiments. Thus, ANO1 could be activated by calcium influx through TRPV3 or TRPV4 in cells co-expressing these ion channels. Therefore, we decided to focus on the interaction between TRPV3 and ANO1 in this study because their interaction had not received attention in the literature.

**TRPV3-ANO1 interaction in HEK293T cells**. We performed whole-cell patch-clamp experiments using HEK293T cells heterologously expressing TRPV3 and ANO1 to investigate their functional interaction. NMDG-Cl bath and pipette solutions were used to identify chloride currents through ANO1 because NMDG is known not to permeate pores of cation channels. We used camphor as a TRPV3 agonist since a previous report showed other TRPV3 agonizts, 2-APB and carvacrol, inhibited ANO1 currents[20]. Under these conditions, chloride currents were clearly observed in cells expressing both human TRPV3 (hTRPV3) and human ANO1 (hANO1), but not in cells expressing hTRPV3 or hANO1 alone (Fig. 2a, b). The currents were interpreted to be chloride currents passing through hANO1 that had been activated by calcium entering cells through hTRPV3. The currents were observed even with intracellular 1, 2-bis (o-aminophenoxy) ethane-*N,N,N',N'*-tetraacetic acid (BAPTA) (5 mM), which is a relatively fast calcium chelator. Because calcium is chelated quickly in the presence of high concentrations of BAPTA, the increases in calcium concentrations occur only in the vicinity of calcium channels[21]. Thus, TRPV3-ANO1 interaction could occur in a local calcium nanodomain. Since previous studies suggested that both TRPV1 and TRPV4 physically interacted with ANO1[2,3,22], we performed immunoprecipitation and Western blotting experiments using anti-ANO1 and anti-TRPV3 antibodies with extracts from HEK293T cells (Fig. 2c and Suppl. Fig. 11). TRPV3 and ANO1 proteins were co-immunoprecipitated in cells expressing both proteins while there were no TRPV3 bands in the extracts from cells transfected with *hANO1* cDNA, *hTRPV3* cDNA, or pcDNA3.1 plasmid alone, indicating the physical interaction of hTRPV3 with hANO1. These results suggested functional and physical interaction between hTRPV3 and hANO1 in the heterologous expression system.

**TRPV3-ANO1 interaction in NHEKs**. Intracellular calcium increases were observed in all NHEKs upon camphor application (Fig. 1c). Therefore, we performed whole-cell patch-clamp experiments in NHEKs. Camphor-induced chloride currents were observed in 148 mM chloride-containing bath solution (Fig. 3a).

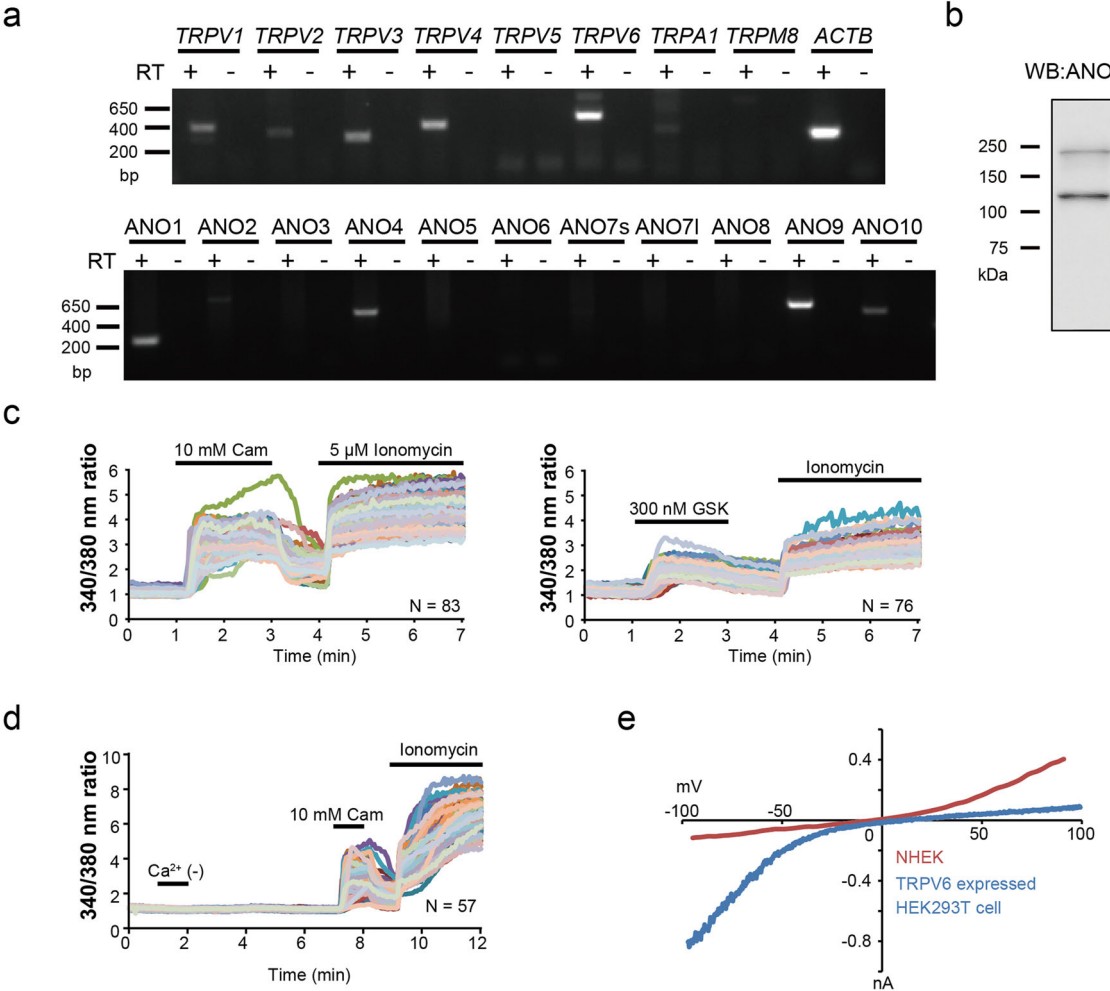

**Fig. 1 TRP channels and ANOs expression in NHEKs. a** RT-PCR of *TRPs* and *ANOs* in NHEKs. Uncropped gel images are shown in Suppl. Fig. 11. **b** Western blot of ANO1 in NHEKs. The predicted band size of ANO1 is 114 kDa. **c, d** Calcium imaging in NHEKs. Cam Camphor, a TRPV3 agonist; GSK GSK1016790A, a TRPV4 agonist; $Ca^{2+}$ (−) Calcium-free bath solution. **e** Comparison of current–voltage relationships. The red line indicates current–voltage relationship of basal current in NHEKs using a standard bath and pipette solution. The blue line indicates current–voltage relationship in TRPV6-expressing HEK293T cells using a standard bath solution and NMDG-Cl pipette solution. The holding potential was −60 mV and ramp-pulses were applied from −100 to +100 mV for 300 ms duration every 5 s.

The reversal potential of the chloride currents was shifted to positive potentials when the extracellular chloride concentration was changed to 4 mM (Fig.3b, c). That result indicated that chloride was a major ion carrier of the camphor-induced currents. In addition, we performed calcium imaging in the calcium-fee bath solution for evaluating the contribution of intracellular calcium store to the camphor-induced intracellular calcium increases (Fig.3d). The release of calcium from the endoplasmic reticulum was not a major contributor to increases in intracellular calcium concentrations in NHEKs because the increases in intracellular calcium concentrations were small in the calcium-free bath solution. This interpretation was confirmed in the patch-clamp experiments in which the camphor-induced currents were very small in the extracellular calcium-free solution (Fig. 3e). Therefore, TRPV3 agonist-induced chloride currents through ANO1 in NHEKs mainly depend on the calcium influx through TRPV3 from extracellular regions. Furthermore, when we examined whether TPRV4 contributes to the camphor-induced currents, we found that 10 mM camphor did not activate hTRPV4 and that HC067047, a selective inhibitor of TRPV4, did not suppress the 10 mM camphor-induced current in NHEK (Suppl. Fig. 3). This ruled out the possibility that camphor nonspecifically activates

TPRV4 and further clarifies the involvement of TRPV3. The camphor-induced chloride currents were inhibited by both Ani9, a strong ANO1 inhibitor (Fig. 3f), and dyclonine, a TRPV3 inhibitor[23,24] (Suppl. Fig. 4). In addition, the camphor-induced currents displayed slow activation kinetics at positive potentials with outwardly rectifying properties and slow recovery after step-pulses, data consistent with ANO1 channels[18] (Suppl. Fig. 5). These results further support TRPV3-ANO1 interaction in NHEKs.

**TRPV3-ANO1 interaction in mouse keratinocytes.** To further confirm the TRPV3-ANO1 interaction, we performed patch-clamp experiments using tail skin keratinocytes from WT and TRPV3$^{-/-}$ mice. We observed camphor-induced large chloride currents in WT keratinocytes, while such currents were never observed in TRPV3-deficient keratinocytes in the NMDG/chloride pipette and bath solutions (Fig. 4a–c). In addition, the camphor-induced currents were inhibited by Ani9 (Fig. 4d). These data clearly support the TRPV3-ANO1 interaction.

**Effects of an ANO1 inhibitor or low chloride medium on NHEK cell migration/proliferation.** Previous studies showed

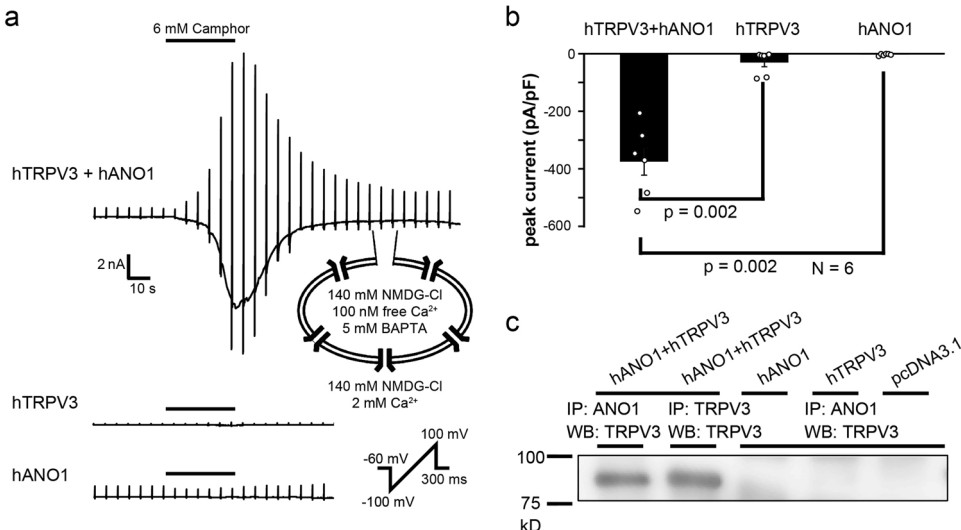

**Fig. 2 TRPV3 and ANO1 interaction in HEK293T cells. a** A representative trace of the camphor-induced currents in HEK293T cells expressing both hTRPV3 and hANO1, hANO1 alone or hTRPV3 alone. All data were collected using an NMDG-Cl bath and pipette solutions. Free calcium in the pipette solution was 100 nM. The holding potential was −60 mV and ramp-pulses were applied from −100 to +100 mV for 300 ms duration every 5 s. **b** Comparison of peak currents of (A) at −60 mV (means ± SEM, N = 6). Statistical significance was determined with a Mann–Whitney test. **c** Immunoprecipitation of ANO1 or TRPV3 and Western blot of TRPV3 in HEK293T cells transfected with *TRPV3* and *ANO1* cDNAs, *Ano1* alone, *TRPV4* alone or pcDNA3.1. Uncropped gel images are shown in Suppl. Fig. 11.

that TRPV3 contributes to itching and warmth sensations and wound healing by keratinocytes[5–8]. It has been strongly suggested that TRPV3 activation accelerates wound healing in the oral cavity[5]. Moreover, the basic histological properties of the oral cavity are similar to those of skin compared to other mucosa in the body[25]. Furthermore, ANO1 could be involved in tissue development after birth[26], and it is well-known to be a positive regulator of migration and proliferation in cancer cells[11,12,15,16]. Therefore, we hypothesized that TRPV3-ANO1 interaction might affect the migration and/or proliferation of NHEKs and the process of wound healing.

To investigate the involvement of ANO1 in wound healing, we analyzed the effects of another ANO1 blocker, T16Ainh-A01 (T16A). The assessment incorporated a culture insert to quantitate cell activity (Fig. 5). In these experiments, NHEKs were cultivated within a culture insert to almost 100% confluency in which cells migrated to spaces between cell clusters[27] (Fig. 5a). NHEKs usually migrated to the open spaces, an area separated by the insert, for ~12 h after the insert was removed. In this way, migration and proliferation filled the area by more than 80% within 24 h. Cell migration and/or proliferation were significantly inhibited by dyclonine, a selective inhibitor of TRPV3 (Suppl. Fig. 6), but not by HC067047, a selective inhibitor of TRPV4 (Suppl. Fig. 7). In addition, cell migration and/or proliferation in T16A (5 μM)-containing medium was obviously inhibited without affecting ANO1 protein levels (Fig. 5b, c and Suppl. Figs. 8, 11). Importantly, the inhibition was lost after the washout of T16A. Those observations suggested that the T16A effect was not due to cell death or irreversible cell damage. Furthermore, Ani9 also inhibited cell migration and/or proliferation (Suppl. Fig. 9). The fact that cell migration and/or proliferation were reduced by both TRPV3 and ANO1 inhibitors, strongly indicated that TRPV3-ANO1 interaction is involved in the migration and/or proliferation of NHEKs. We analyzed cell migration velocity using time-lapse imaging with a confocal microscope and cell proliferation using an MTT assay (Fig. 5d–f). Migration velocity was reduced after T16A application, and the reduction lasted throughout the inhibition of ANO1 (Fig. 5d). Moreover, the migration velocity recovered to the initial level after the washout

of T16A (Fig. 5d, e). Cell proliferation was also reduced by T16A application (Fig. 5f). These results suggested the importance of chloride ions for cell migration and proliferation. Therefore, we performed an assay with a culture insert in a low chloride medium (Fig. 6). Intracellular chloride concentrations should be reduced upon depletion of extracellular chloride[28]. After the removal of the culture inserts, the filled areas were drastically reduced in the low chloride-containing medium, an effect that was lost after the change back to the control medium (Fig. 6). These results indicated that chloride flux through ANO1 plays critical roles in cell migration and/or proliferation.

**Direction of chloride movement through chloride channels in NHEKs.** Although the previous results suggested the importance of chloride ions for cell migration and/or proliferation, the actual roles of chloride ions in keratinocytes are largely unknown. To address this question, we attempted to determine the direction of chloride movement. Chloride permeation through chloride channels depends on intracellular chloride concentrations and membrane potentials. Although chloride channel function could affect intracellular chloride concentrations, they should be maintained by the function of several chloride transporters[29]. Therefore, we examined the expression patterns of chloride transporters, including Na-K-Cl cotransporters (NKCCs) and K-Cl cotransporters (KCCs), using RT-PCR. mRNA expression of the genes coding for NKCC1, KCC1, KCC2, KCC3, and KCC4 was suggested (Fig. 7a and Suppl. Fig. 11). KCC2 is a neuron-specific KCC, and intracellular chloride concentrations are kept at a low level through chloride efflux by KCC2 in cells in the central nervous system, and opening of the chloride-permeable channels causes chloride influx. Therefore, we performed chloride-imaging experiments using the chloride indicator, MQAE[30–32]. The calculated intracellular chloride concentrations of NHEKs were relatively low (6.8 ± 1.3 mM) (Fig. 7b, c), which is consistent with KCC2 expression at least at the mRNA level in NHEKs. The calculated equilibrium potential for chloride ions (−75.7 mV) suggests that chloride influx through ANO1 would occur at the reported resting membrane potentials in NHEKs (−24 to −40 mV)[33–35].

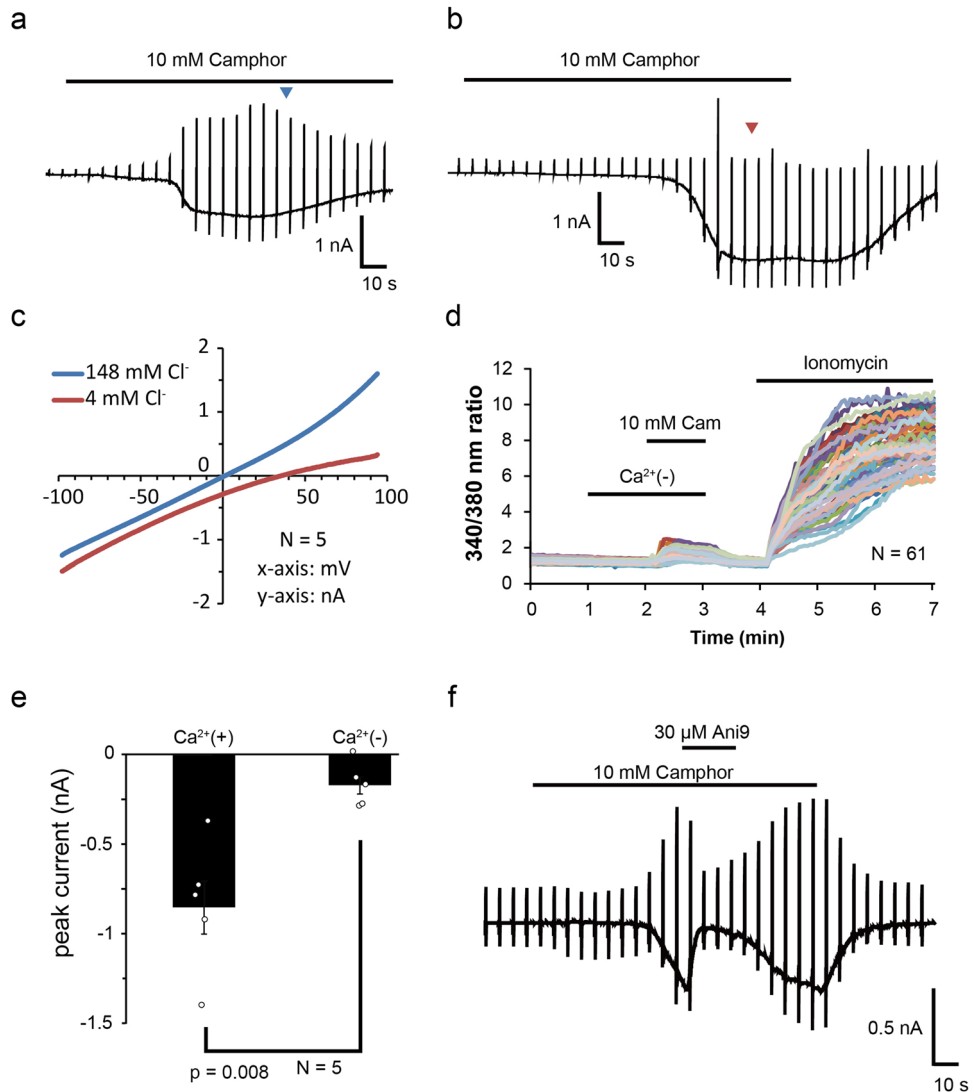

**Fig. 3 TRPV3 and ANO1 interaction in NHEKs. a**, **b** Representative traces of camphor-induced currents in NHEKs using an NMDG-Cl bath solution containing 148 mM chloride (**a**) or an NMDG-aspartate bath solution containing 4 mM chloride (**b**). The pipette solution contained 140 mM NMDG-Cl and 100 nM free calcium. The holding potential was −60 mV and ramp-pulses were applied from −100 to +100 mV for 300 ms duration every 5 s. **c** Comparison of current–voltage relationships of the currents at arrowheads in (**a**, **b**). **d** Calcium imaging of NHEKs upon camphor application with and without (Ca$^{2+}$ (−)) extracellular calcium. **e** Comparison of camphor-induced peak currents at −60 mV in an NMDG-Cl bath solution with 2 mM (Ca$^{2+}$ (+)) or without (Ca$^{2+}$ (−)) extracellular calcium (means ± SEM, $N = 5$). Statistical significance was determined with a Mann–Whitney test. **f** Camphor-induced current with an ANO1 inhibitor, Ani9, using an NMDG-Cl bath solution containing 148 mM chloride.

**An ANO1 inhibitor induces MAP kinase phosphorylation.** Previous studies suggested that low intracellular chloride concentrations induce the phosphorylation of mitogen-activated protein kinase (MAPK), although its precise mechanisms are not well known. MAPK cascades are involved in the life and death of many cells[36,37] (Fig. 8a). For instance, extracellular signal-related kinase (ERK), which is phosphorylated by MAPK kinase (MKK) 1/2, is involved in cell proliferation and differentiation. On the other hand, p38 and c-Jun N-terminal kinase (JNK), which are phosphorylated by MKK3/4/6 and MKK4/7, respectively, induce cell cycle arrest and apoptosis. Hence, we investigated MAPK phosphorylation using a Western blot method (Fig. 8b, c and Suppl. Fig. 11). An ANO1 inhibitor, T16A, increased the phosphorylation of p38, but not that of ERK or JNK. These results suggested that ANO1 is involved in cell cycle arrest and/or apoptosis. However, ANO1 inhibition did not induce cell death in the culture insert assay based upon the fact that cell appearance visualized with a calcein-AM staining was not affected by ANO1

inhibition (Fig. 5). In addition, there were no effects of T16A treatment on the expression of differentiation-related genes, including *KRT1*, *IVL*, and *TGM1*, in both differentiated and undifferentiated conditions (Suppl. Figs. 10, 11). This result is consistent with the lack of effects of the T16A treatment on ERK phosphorylation (Fig. 8b), which is known to be related to differentiation (Fig. 8a). Therefore, we decided to focus on cell cycle arrest.

**An ANO1 inhibitor-induced cell cycle arrest during the culture insert assay.** Because the MAPK analyses suggested cell cycle arrest by ANO1 inhibition, we performed a cell cycle assay by using a redox dye (Fig. 9). Redox conditions are closely related to the cell cycle[38]. For instance, intracellular redox conditions in cells in the G0/G1 phases are relatively reductive, whereas redox conditions are gradually shifted to more oxidative ones upon progression to the G2/M phases. In this assay system, cells in G0/G1 phase, S phase, and G2/M phases can be visualized as yellow-

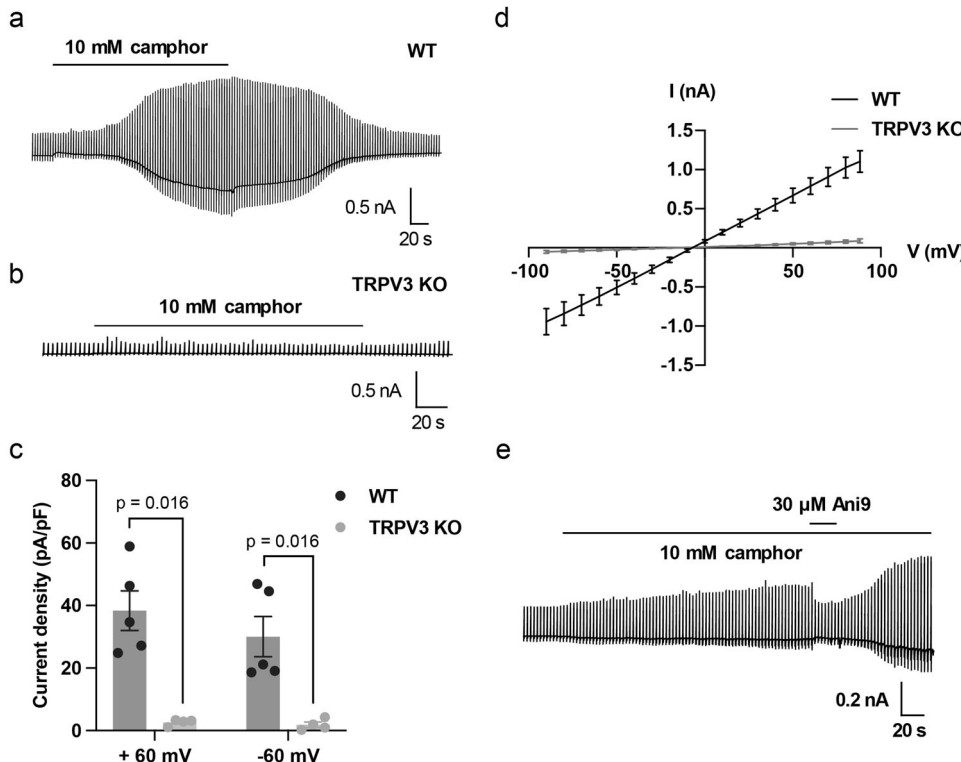

**Fig. 4 TRPV3 and ANO1 interaction in mouse primary skin keratinocytes. a, b** Representative traces of 10 mM camphor-induced currents in primary keratinocytes isolated from WT (**a**) and TRPV3$^{-/-}$ (**b**) mice. All the recordings were performed with standard NMDG-Cl bath and pipette solutions (with 100 nM free calcium in the pipette). The holding potential was −60 mV and ramp-pulses were applied from −100 to +100 mV for 300 ms duration every 3 s. **c** Comparison of current–voltage relationships. N equals 5 for WT (black) and 4 for TRPV3$^{-/-}$ (gray) keratinocytes. Error bars indicate SEM. **d** Comparison of densities of the camphor-induced currents in WT (black) and TRPV3$^{-/-}$ (gray) keratinocytes at +60 and −60 mV (means ± SEM). Statistical significance was determined with a Mann–Whitney test. **e** A representative trace of 10 mM camphor-induced currents with Ani9 inhibition in a WT keratinocyte. A representative trace of 10 mM camphor-induced currents with Ani9 inhibition in a WT keratinocyte. The holding potential was −60 mV and ramp-pulses were applied from −100 to +100 mV for 300 ms duration every 3 s.

green, green, and dark blue, respectively (Fig. 9a). To clarify the color variation, each cell was visualized as a red color depending on signal levels (Fig. 9b). T16A treatment increased cell populations in G0/G1 phases and reduced cell populations in S phase during the culture insert assay (Fig. 9b, c). This result indicated that cell cycle progression from G0/G1 to the S phase was suppressed by ANO1 inhibition.

## Discussion

In contrast to previous studies, our results showed that functional expression of TRP channels was limited to TRPV3 and TRPV4, although mRNA expression of a number of other TRP channels was observed in NHEKs (Fig. 1 and Suppl. Fig. 2). Protein expression reportedly changes depending on differentiation conditions in keratinocytes[39]. Thus, TRP channels whose functional expression was not confirmed in our experiments could be functional under other specific differentiation conditions. In particular, TRPV6 and TRPC6 were reported to be involved in keratinocyte differentiation, and TRPC6 expression was increased by differentiation stimuli[40,41]. We observed TRPV3-ANO1 interaction in this study, and our laboratory reported similar results for TRPV4-ANO1 in epithelial cells[2,3]. However, the interaction between ANO1 and other TRP channels whose functional expression was not confirmed in this study might occur in differentiated cells.

TRPV4 is well known to interact with ANO1, and TRPV3 was found to be a new candidate partner of ANO1 in NHEKs. Patch-clamp experiments showed TRPV3-ANO1 interaction in

HEK293T cells and NHEKs (Figs. 2, 3 and Suppl. Fig. 11). In addition, TRPV3 was co-immunoprecipitated using an anti-ANO1 antibody in HEK293T cells expressing both TRPV3 and ANO1. These results suggest that TRPV3 and ANO1 form a complex with a calcium nanodomain and that ANO1 is effectively activated by calcium entering cells through TRPV3, as is the case with TRPV4. It is known that increases in intracellular calcium concentrations downstream of activation of calcium-sensing receptor (CaSR) induce differentiation in keratinocytes[42]. Thus, there could be two different types of increases in intracellular calcium concentrations: a local one through TRPV3 activation that activates ANO1 and a global one that can be caused by several mechanisms such as CaSR signaling.

We assessed the physiological function of TRPV3-ANO1 interaction. A previous study showed that TRPV3 was involved in wound healing in keratinocytes[5]. In fact, TRPV3 is sensitized by epidermal growth factor receptor (EGFR) following stimulation by EGF released during wound healing. In addition, transforming growth factor alpha (TGFα), an EGFR ligand, is released from keratinocytes by TRPV3 activation. This autocrine system is thought to represent one of the molecular mechanisms for wound healing via TRPV3 activity. Our assays with culture inserts indicate that inhibition of either TRPV3 or ANO1 reduces cell migration and/or proliferation (Fig. 5 and Suppl. Figs. 6, 9). Thus, TRPV3-ANO1 interaction could be an important molecular mechanism for wound healing in the skin, suggesting that there are two processes for wound healing that involve TRPV3 activity. A previous report suggested that membrane localization of ANO1 is important for

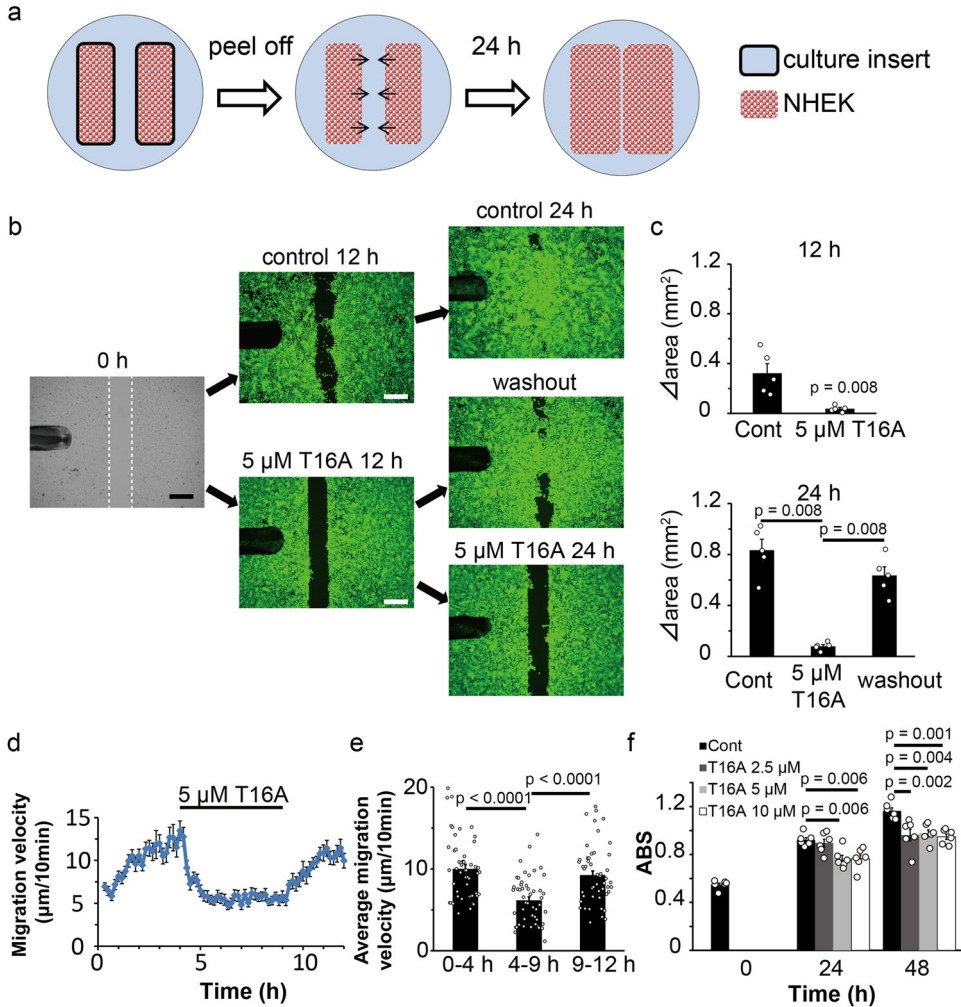

**Fig. 5 Effects of an ANO1 inhibitor on cell migration/proliferation in the culture insert assay. a** Schematic image of culture insert assay. **b** Culture insert assay in medium with or without 5 μM T16A. Bright-field at 0 h and calcein staining at 24 h. White dotted lines at 0 h indicate the borders of the cells. Washout indicates the change of medium from T16A-containing medium to control medium at 12 h. Scale bars indicate 500 μm. **c** Measurements of increased areas (Δ area) at 12 or 24 h in the medium with or without 5 μM T16A. Data represent means ± SEM ($N = 5$). Statistical significance was determined with a Mann–Whitney test. **d** Migration velocity of NHEKs with or without 5 μM T16A in culture insert assay. Data represent means ± SEM ($N = 50$). **e** Average migration velocity from (**d**). Each column indicates the average velocity during the indicated time period. Data represent means ± SEM ($N = 50$). Statistical significance was determined with a Mann–Whitney test. **f** MTT assay of NHEKs cultured in the indicated medium. ABS indicates absorbance. Data represent means ± SEM ($N = 6$). Statistical significance was determined with the one-way ANOVA followed by Dunnett correction.

cancer cell migration independent of chloride channel activity[43]. In that experiment, only CaCCinh-A01 decreased cell proliferation and induced ANO1 protein degradation among several ANO1 inhibitors (including T16A) that they used. However, in our experiments, the T16A treatment decreased NHEK proliferation without effects on ANO1 protein levels (Fig. 5 and Suppl. Figs. 8, 11). Moreover, our finding that low chloride conditions also inhibited cell migration and/or proliferation in culture insert assays (Fig. 6) supports the importance of chloride movement through ANO1 for wound healing.

Another report revealed that ANO1 induces hyperproliferation of HaCaT cells, a special cell line of keratinocytes[14], a result that is consistent with our results. However, the authors showed that inhibition of ANO1 reduced the phosphorylation of ERK, one of the MAPKs, and they did not discuss the involvement of p38. In our experiments, ANO1 inhibition did not affect ERK phosphorylation, while p38 phosphorylation was induced. This difference could be due to the fact that HaCaT is a cell line with a special calcium requirement that differs from normal keratinocytes.

The direction of chloride movement is strictly regulated by the balance of intracellular chloride concentrations and membrane potentials. For example, intracellular chloride concentrations are maintained at a low level by KCC2 activity in mature neurons of the central nervous system[29], leading to hyperpolarization upon chloride influx. We showed that the basal intracellular chloride concentrations in keratinocytes were low, consistent with the expression of KCC2 (Fig. 7 and Suppl. Fig. 11). Because the resting membrane potentials of skin keratinocytes are reported to range from −24 to −40 mV[33–35], ANO1 opening should induce chloride influx and its inhibition likely decreases intracellular chloride ions within keratinocytes. Although KCC2 is generally thought to be neuron-specific[44], a recent report showed that pancreatic β cells also expressed KCC2[45], suggesting that KCC2 might be more widely expressed than expected. Since both keratinocytes and neurons are ectodermal cells in origin[46,47], it would be reasonable that neuron-specific KCC2 could control intracellular chloride concentrations in both neurons and keratinocytes.

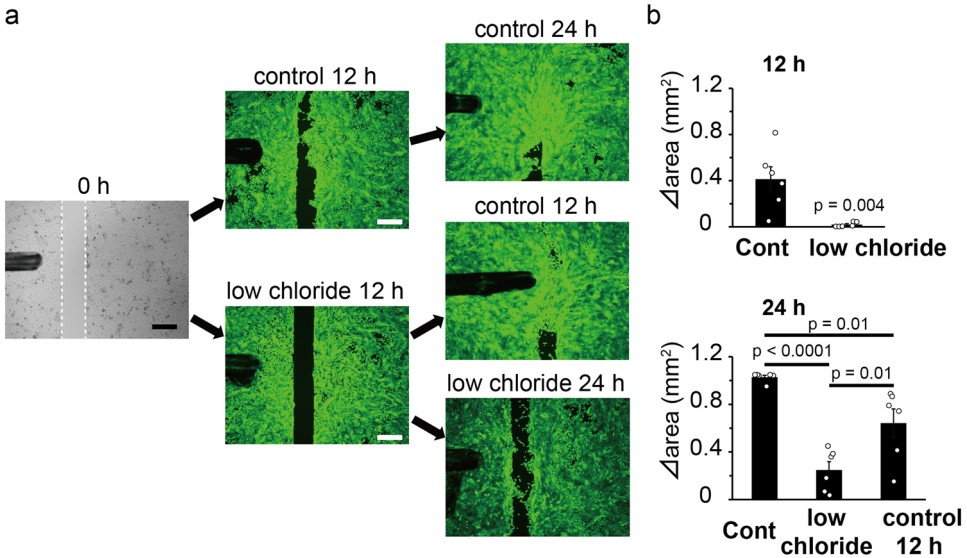

**Fig. 6 Effects of a low chloride medium on cell migration/proliferation in the culture insert assay. a** Culture insert assay in a low chloride medium or control medium. Bright fields at 0 h and calcein staining at 24 h. White dotted lines at 0 h indicate the borders of the cells. Control 12 h indicates a change of medium from low chloride medium to control medium at 12 h. Scale bars indicate 500 μm. **b** Measurements of increased areas (Δ area) at 12 or 24 h in the low chloride medium or control medium. Data represent means ± S.E.M ($N = 6$). Statistical significance was determined with the Mann–Whitney test or one-way ANOVA followed by Tukey correction.

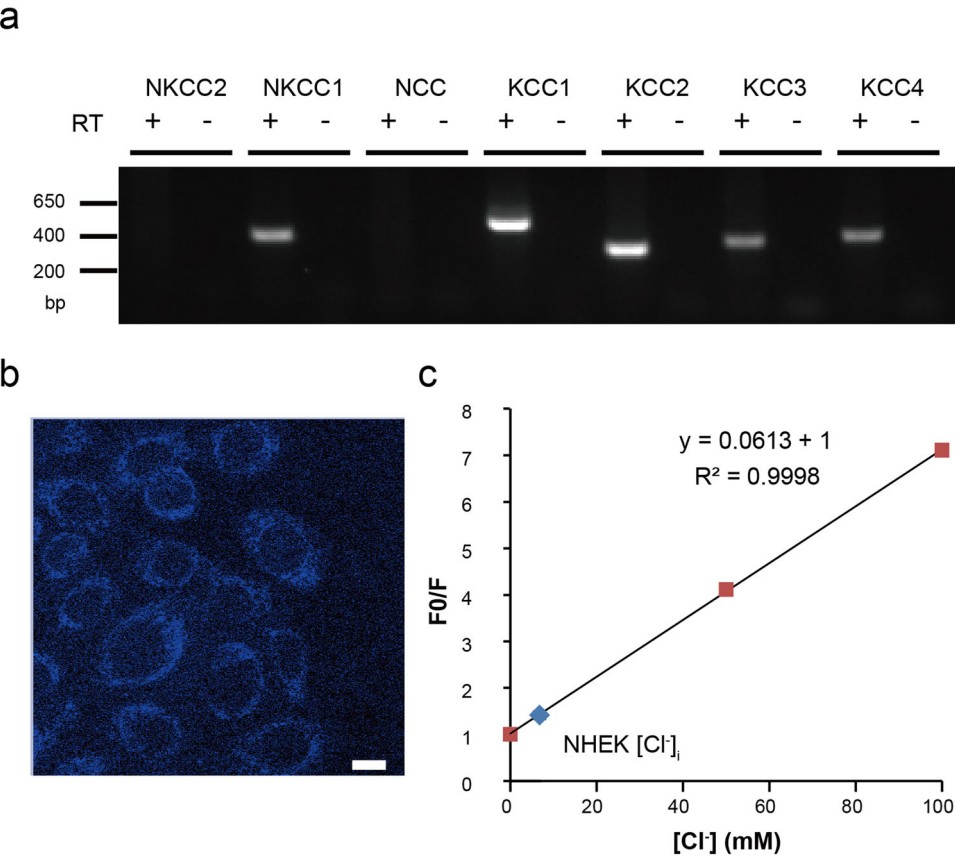

**Fig. 7 Expression of cation-chloride cotransporter genes and calculated chloride concentrations in NHEKs. a** RT-PCR assessment of cation-chloride cotransporter genes in NHEKs. RT indicates reverse transcription. Uncropped gel images are shown in Suppl. Fig. 11. **b** A representative image of chloride ion-quenched fluorescent indicator, MQAE, in NHEKs. The scale bar indicates 10 μm. **c** A calibration curve and calculated intracellular chloride concentrations in NHEKs. The calibration curve was made with a Stern–Volmer plot, $F_0/F = 1 + K_q[Cl]$. $F_0$: fluorescence intensity at 0 mM chloride, F fluorescence intensity at each chloride concentration, $K_q$ extinction coefficient.

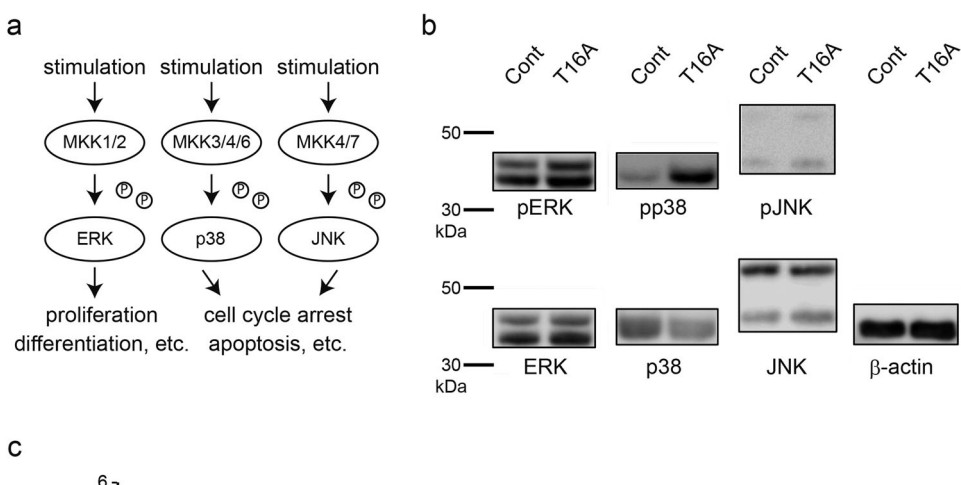

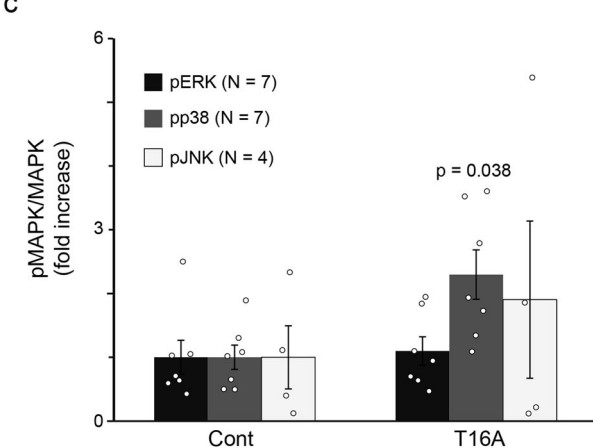

**Fig. 8 Effects of an ANO1 inhibitor on MAP kinase phosphorylation. a** Schematic representation of MAP kinase signaling. **b** Representative images of Western blotting for total MAP kinases and phosphorylated MAP kinases. Uncropped gel images are shown in Suppl. Fig. 11. **c** Quantitative analysis of Western blotting for phosphorylated/total MAP kinases. Data represent means ± SEM ($N = 7$ for pERK and pp38, $N = 4$ for pJNK. Statistical significance was determined with a Mann–Whitney test.

In cancer cells, decreases in intracellular chloride concentrations induce phosphorylation of p38 and JNK. That change, in turn, increases p21 expression, leading to the inhibition of CDK2/Cyclin E complex function, followed by the arrest of the cell cycle in the G1 phase[48,49]. Consistent with previous reports in cancer cells, p38 phosphorylation was induced by ANO1 inhibition in NHEKs (Fig. 8 and Suppl. Fig. 11). p38 is known to negatively regulate the cell cycle[36]. That finding, together with the reduction in cell proliferation achieved by ANO1 inhibition (Fig. 5f), suggests that ANO1 inhibition causes cell cycle arrest. As expected, the T16A treatment increased the proportion of cells within the $G_0/G_1$ phases and reduced cells within the S phase in the culture insert assay (Fig. 9b, c). The results indicate that cell cycle arrest occurs between the $G_0/G_1$ phases and the S phase. That finding is consistent with previous studies that showed that cells were arrested at the G1/S checkpoint when intracellular chloride concentrations were reduced in cancer cells[48,49]. Thus, these observations suggest that the cell cycle of keratinocytes is also controlled by changes in intracellular chloride concentrations downstream of TRPV3-ANO1 signaling.

Our study suggests that TRPV3 induces chloride influx through ANO1. This chloride influx could properly maintain the cell cycle through the inhibition of p38 phosphorylation. Thus, the decreases in cell proliferation caused by T16A (Fig. 5f) could be partly explained by cell cycle arrest through ANO1 inhibition. Our data also showed that ANO1 inhibition slowed cell migration (Fig. 5d, e). Although the mechanisms for controlling migration velocity by ANO1 are unknown, a previous report showed that

molecules inhibiting the cell cycle, such as p21 rearranged the cytoskeleton through a ROS-mediated pathway, thereby regulating cell migration[50]. Thus, chloride could also control cell migration through the same molecules that inhibit the cell cycle. While the exact mechanism by which chloride regulates MAPK phosphorylation is unknown, intracellular chloride concentrations could directly regulate phosphorylation levels as reported for the regulation of phosphatase activity[51]. Further studies are definitely necessary to clarify the roles of chloride in the cell signaling pathways of keratinocytes. Nonetheless, the interaction of TRPV3 and ANO1 found in this study suggests that they could positively regulate keratinocyte proliferation and migration during wound healing.

This study demonstrates that chloride ion is a regulator of the cell cycle in keratinocytes. These data suggest new approaches to promoting wound healing. Keratinocyte hyperproliferation is the cause of acne at the follicular site. Normalizing this hyperproliferation is an important therapeutic strategy for acne[52]. However, the importance of chloride ions in such a therapeutic setting has not been emphasized. Our analysis of TRPV3-ANO1 interaction represents a promising starting point for future investigations into the detailed mechanisms underlying intracellular chloride concentrations. Thus, our study could shed light on the importance of chloride ions in skin homeostasis.

## Methods

**Animals.** Homozygous TRPV3-deficient (TRPV3$^{-/-}$) mice were provided by Dr. Patapoutian. Wild-type (WT) and TRPV3$^{-/-}$ mice of C57BL/6 background were

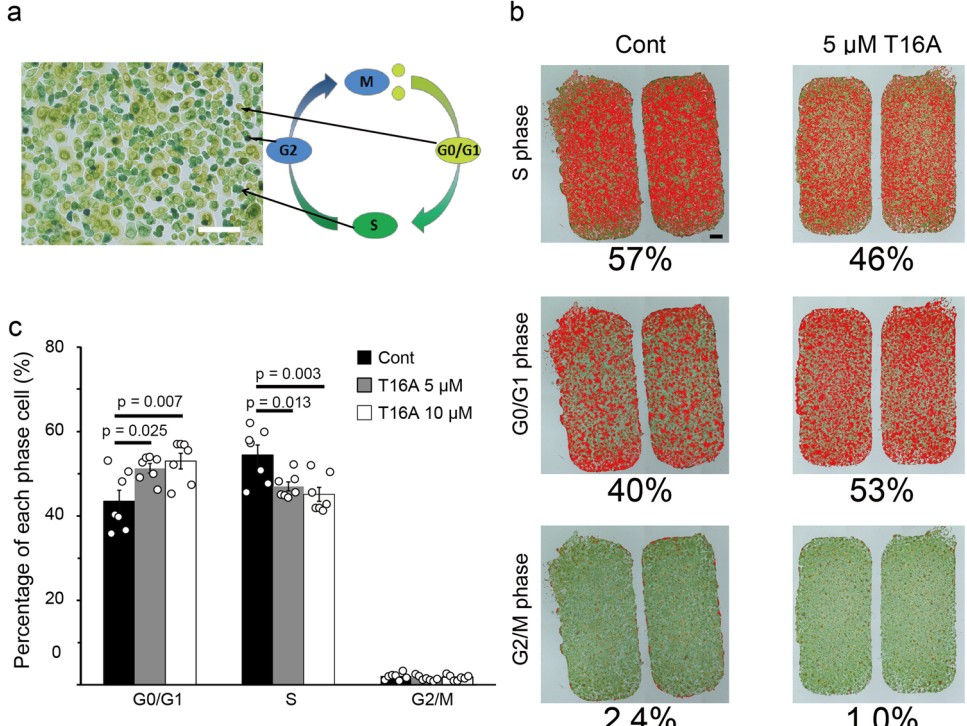

**Fig. 9 Effects of an ANO1 inhibitor on the cell cycle of NHEKs. a** A representative image of redox dye staining for cell cycle analysis. The scale bar indicates 100 μm. **b** Representative images of cells in the presence or absence of 5 μM T16A in the cell culture insert. Cells marked in red are in the indicated phase. The scale bar indicates 500 μm. **c** Comparison of the percentages of each phase. Data represent means ± SEM ($N = 7$). Statistical significance was determined with one-way ANOVA followed by Dunnett correction.

maintained under SPF conditions in a controlled environment (12-h light/dark cycle with free access to food and water, 25 °C, and 50–60% humidity). Seven weeks old male mice were used. All procedures were approved by the Institutional Animal Care and Use Committee of the National Institute of Natural Sciences and carried out according to the National Institutes of Health and National Institute for Physiological Sciences guidelines.

**Reagents and antibodies**. T16Ainh-A01 (T16A, Calbiochem) and Ani9 (Sigma-Aldrich) were used as an Ano1 inhibitor. Dyclonine (MedChemExpress) was used as a TRPV3 inhibitor. The following antibodies were used: rabbit anti-ANO1 antibody (Abcam, ab53213, 1:5 for Western blotting), (Abcam, ab53212, 1:100 for immunoprecipitation), rabbit anti-phospho-ERK (extracellular signal-related kinase) antibody (Cell Signaling Technology, #4370, 1:1000), rabbit anti-phospho-p38 antibody (Cell Signaling Technology, #4511, 1:1000), rabbit anti-phospho-JNK (c-Jun N-terminal Kinase) antibody (Cell Signaling Technology, #4668, 1:1000), rabbit anti-ERK antibody (Cell Signaling Technology, #4695, 1:1000), rabbit anti-p38 antibody (Cell Signaling Technology, #8690, 1:1000), rabbit anti-JNK antibody (Cell Signaling Technology, #9252, 1:1000), mouse anti-β-actin antibody (Abcam, ab6276, 1:2500), rabbit anti-TRPV3 antibody (Cell Signaling Technology, #3484, 1:1000 for Western blotting, 1:50 for immunoprecipitation), anti-rabbit-HRP antibody (Cell Signaling Technology, #7074, 1:2000) and anti-mouse-HRP antibody (Cell Signaling Technology, #7076, 1:2000).

**Cell culture**. HEK293T cells (ATCC CRT-3216) were maintained at 37 °C in 5% $CO_2$ in Dulbecco's modified Eagle's medium (Wako) containing 10% fetal bovine serum (BioWest), 50 units/mL penicillin, 50 μg/mL streptomycin (Life Technologies), and 2 mM/L glutamine (GlutaMAX, Life Technologies). Normal human epidermal keratinocytes (NHEK, Adult, KURABO) were maintained at 37 °C in 5% $CO_2$ in Humedia-KG2 (KURABO). Custom MCDB153 medium lacking NaCl (Research Institute for Functional Peptides) was used for low chloride experiments. Custom MCDB153 medium was used by adding 130 mM NaCl or 130 mM sodium aspartate and 0.1 mM O-phosphorylethanolamine (Sigma), 0.1 mM ethanolamine (Sigma), 0.5 μg/mL hydrocortisone (Sigma), 5 ng/mL epidermal growth factor (EGF, Miltenyi Biotec), and 5 μg/mL insulin (Sigma).

**Primary mouse keratinocyte isolation and culture**. Skin keratinocytes were isolated based on the previous description with minor modifications[53]. Briefly, mice tails were used for preparing the keratinocytes. The dissociated tails were incubated overnight at 4 °C in 4 mg/mL DISPASE II (Wako, 383-02281) in customized MCDB153 medium (CSR) containing 5 ug/mL insulin (Sigma, I1882),

0.4 ug/mL hydrocortisone (Sigma, H0888), 10 ug/mL transferrin (Sigma, T8158), 14.1 ug/mL phosphorylethanolamine (Sigma, P0503), 10 ng/mL epidermal growth factor (Sigma, E4127), 25 ug/mL gentamicin (Gibco, 15710064), 50 units/mL penicillin, 50 ug/mL streptomycin and 40 ug/mL bovine pituitary extracts (Kyokuto, 20200). After 16 h of incubation, the epidermis was detached and incubated with 0.25% trypsin (Gibco, 15050065) for 20 min at room temperature with the basal layer down. Keratinocytes were next mechanically dissociated with dissecting forceps and filtered through a 100-um cell strainer. Cells were next seeded on coverslips for patch-clamp experiments. All recordings were performed on day 3 and day 4 after isolation.

**RT-PCR**. Total RNA was purified from NHEKs using Sepasol-RNA I Super G (Nacalai Tesque) or RNeasy Micro (QIAGEN). Reverse transcription was performed using Super Script III reverse transcriptase (Invitrogen) for 50 min at 50 °C. For investigation of mRNA expression of transient receptor potentials (TRPs), anoctamins (ANOs), and Cation-Chloride-Cotransporters (CCCs) in NHEKs, DNA fragments were amplified using EmeraldAmp PCR Master Mix (TAKARA) with PCR primers shown in Table 1. The PCR products were confirmed by electrophoresis on a 2% agarose gel containing ethidium bromide.

**Western blotting**. Proteins were extracted from NHEKs treated with 10 μM T16A or control medium for 13 h. The cells were washed with cold PBS and lysed by treatment with lysis buffer (1% Triton X-100 contained 150 mM NaCl, 10 mM Tris-HCl, 1 mM EDTA, 1 mM $Na_3VO_4$, and protease inhibitor cocktail, cOmplete (Roche), pH 7.5). Following centrifugation at 10,000×g for 30 s, the supernatants were denatured by treatment with SDS buffer containing 0.5 M Tris-HCl, 10% sodium dodecyl sulfate, 6% β-mercaptoethanol, 10% glycerol, 0.01% bromophenol blue, 100 mM dithiothreitol, at 90 °C for 5 min. The protein samples were used in SDS-PAGE.

**Transfection**. Transient transfection of HEK293 cells was achieved with Lipofectamine Transfection Reagent (Life Technologies), PLUS Reagent (Life Technologies), and Opti-MEM I Reduced Serum Medium (Life Technologies). Plasmid DNAs (hTRPV6/pcDNA3.1, hTRPV3/pcDNA3, hANO1/pcDNA3.1, or pcDNA3.1) were transfected with pGreen Lantern 1 into HEK293T cells, and the transfected cells were used for patch-clamp recording and immunoprecipitation 16–30 h after transfection.

**Table 1 Primer list for reverse transcription-PCR (RT-PCR).**

| | Forward primer | Reverse primer |
|---|---|---|
| *TRPV1* | CTGCGGACCCACTCCAAAAGGA | AGAGCAGCAGGCTCTCCAGATC |
| *TRPV2* | CTGCACATCGCCATTGAGAAGA | TTGGAGGAGCCCATCATACATG |
| *TRPV3* | GCTGAAGAAGCGCATCTTTGCA | TCATAGGCCTCCTCTGTGTACT |
| *TRPV4* | TACCTGTGTGCCATGGTCATCT | TGCTATAGGTCCCCGTCAGCTT |
| *TRPV5* | GTCCCAGCCCCAAATAGACC | GCAGTCTGACCTGCAAAAGC |
| *TRPV6* | GTAGAAGTGGCCTAGCTCCTCGG | AGCCTACATGACCCCTAAGGACG |
| *TRPA1* | GACCACAATGGCTGGACAGCT | GTACCATTGCGTTGAGGGCTGT |
| *TRPM8* | CCTGTTCCTCTTTGCGGTGTGGAT | TCCTCTGAGGTGTCGTTGGCTTT |
| *β-actin* | GATCCTCACCGAGCGCGGCTACA | GCGGATGTCCACGTCACACTTCA |
| *ANO1* | GAGATCGGTTCCCAGCCTAC | GGGACCTCGATCTTGGTGAG |
| *ANO2* | TTCATGGCTCTGTGGGCTAC | GCAGACTGGTTGCTCTCCTT |
| *ANO3* | CGTAGGCCACCCAGGAAAAT | CCAGTTCTGGATCAACGGGT |
| *ANO4* | GAGAAGCTGGGCAAAAGTGC | GTGCTTTTGCACAGCCAGAA |
| *ANO5* | GAGTGAGAGCCTGAGCAGC | TTCCGCCTTTAACTCTGCGT |
| *ANO6* | CGAACCCCGGAGTTTGAAGA | CTCCCATGGTGCGTGTACTT |
| *ANO7s* | CAAGCCCCGGATCGACTTC | CATGACAGCCTCCACGTACC |
| *ANO7l* | AGGAGCCGGGAAGCCTTG | TGGTTGGGTAACTCCTGCAA |
| *ANO8* | CGGGCCGTCCGTAGC | TCATCGGTCGTGTCTGGGAAG |
| *ANO9* | TCTTTGGGATCCGTGCTGAC | GGGTGAGCTTGGCAAAAGTG |
| *ANO10* | TCTTTCACACCTTTGGTGGTCA | TGCCAGACGTGAGCAATCTT |
| NKCC2 | GGCTCCATCACAGTGGTGAT | TGGGGATCCTCCAAATCTCCT |
| NKCC1 | TTTGCAGAAACCGTGGTGGA | GCTGACTGAGGATCTGCAAG |
| NCC | TTTGACGATGGAGGCCTCAC | CTGCCGAAGGGACTTGACTC |
| KCC1 | CGGCAGATGAGACTGACCAA | CCAGGCCACAAGATGACACT |
| KCC2 | AAGCCGGACCAATCCAATGT | CTGTTACCCCAGACCACAGC |
| KCC3 | GCACTCTTTGAGGAAGAAATGGAC | GCTGGCACCACTCCATTAGT |
| KCC4 | CGTTTCCGCAAAACCAGGAG | TTCCTGTCGTGGATCAGCTG |
| *KRT1* | ATCAACTACCAGCGCAGGAC | CTCCAGGAACCTCACCTTGTC |
| *IVL* | GCCTTACTGTGAGTCTGGTTGAC | AGCTGCTGATCCCTTTGTGT |
| *TGM1* | GATCGCATCACCCTTGAGTTAC | TCCTCATGGTCCACGTACACAAT |

All primers span an exon-exon junction.

**Calcium imaging**. NHEKs on coverslips were incubated at 37 °C for 30 min in Humedia-KG2 containing 5 μM Fura-2-acetoxymethyl ester (Molecular Probes). The coverslips were washed with a standard bath solution containing 140 mM NaCl, 5 mM KCl, 2 mM CaCl₂, 2 mM MgCl₂, 10 mM HEPES, and 10 mM D-glucose at pH 7.4, adjusted with NaOH. A calcium-free bath solution was prepared by omitting 2 mM CaCl₂ from the standard bath solution and adding 5 mM EGTA. Fura-2 was excited with 340- and 380-nm wavelength lights and the emission was monitored at 510 nm with a CCD camera, Cool Snap ES (Roper Scientific/Photometrics) at room temperature. Data acquired using IP lab software (Scanalytics) and analyzed with ImageJ software (National Institutes of Health). Ionomycin (5 μM, Sigma-Aldrich) was applied to confirm the maximal response of each cell. The high K⁺ bath solution contained 65 mM NaCl, 80 mM KCl, 2 mM CaCl₂, 2 mM MgCl₂, 10 mM HEPES, and 10 mM D-glucose at pH 7.4, adjusted with NaOH.

**Whole-cell patch-clamp**. Transfected HEK293T cells, NHEKs, or isolated mouse keratinocytes were used for whole-cell recordings. Patch pipettes were made from borosilicate glass (type 8250, King Precision Glass) with a five-step protocol using a P-2000 (Sutter Instrument). The pipette resistance was 3–8 MΩ. Currents were recorded at 10 kHz using an Axopatch 200B amplifier (Molecular Devices) and filtered at 5 kHz with a low-pass filter. Currents were digitized with a Digidata 1440 A or 1550 (Axon Instruments). Data acquisition was achieved with pCLAMP 10 software (Axon Instruments). Four extracellular solutions for whole-cell recording were as follows: (1) a standard bath solution (140 mM NaCl, 5 mM KCl, 2 mM CaCl₂, 2 mM MgCl₂, 10 mM HEPES, and 10 mM D-glucose at pH 7.4, adjusted with NaOH); (2) an NMDG-Cl bath solution (140 mM NMDG, 140 mM HCl, 2 mM CaCl₂, 2 mM MgCl₂, 10 mM HEPES, and 10 mM D-glucose at pH 7.4, adjusted with HCl); (3) a calcium-free NMDG-Cl bath solution that was prepared by omitting 2 mM CaCl₂ from the NMDG-Cl bath solution and adding 5 mM EGTA, and (4) an NMDG-aspartate bath solution that was prepared by using L-aspartic acid instead of HCl. The pipette solutions were as follows: (1) a standard pipette solution (140 mM KCl, 5 mM EGTA, 2 mM MgCl₂, and 10 mM HEPES at pH 7.3, adjusted with KOH) or (2) an NMDG-Cl pipette solution (140 mM NMDG, 140 mM HCl, 5 mM BAPTA, 2 mM MgCl₂, and 10 mM HEPES at pH 7.3, adjusted with HCl). CaCl₂ was added to the pipette solution so that the free calcium concentration was 100 or 500 nM. Free calcium concentrations were calculated with the MAXC program at Stanford University.

**Immunoprecipitation**. Proteins were extracted from HEK293T cells after transfection. The cells were lysed as described in *Western blotting*. Following centrifugation at 16,100×*g* for 30 min, the supernatants were incubated in a rotator for 2 h with protein G Mag Sepharose (GE Healthcare). After the removal of magnetic beads, the supernatants were incubated in a rotator overnight with an anti-TRPV3 antibody or anti-ANO1 antibody. After incubation, protein G Mag Sepharose was added, and the solutions were incubated in a rotator for 2 h. After incubation, the magnetic beads were rinsed with washing buffer (50 mM Tris-HCl, 150 mM NaCl, pH 7.5). The proteins were denatured by SDS buffer at 95 °C for 5 min. The protein samples were assessed by SDS-PAGE. Blotting was done with anti-TRPV3 and anti-rabbit-HRP antibodies.

**Culture insert assay**. NHEKs were seeded confluently in two-well culture inserts (ibidi) on glass-bottom dishes (Matsunami). Culture inserts were removed after overnight incubation, followed by washing with PBS. Cells were then cultured in Humedia-KG2, MCDB153 medium, or low chloride MCDB153 medium. After 12 or 24 h of cultivation, calcein-AM (Dojindo) was added to the culture medium to visualize the cells. ImageJ software was used for data analysis. For time-lapse analysis, cells were cultured in a Stage Top Incubator (TOKAI HIT) on a confocal laser scanning microscope (IX83 Olympus) and images were captured every 10 min.

**MTT assay**. NHEKs were seeded on 96-well plates (Falcon). Cells were cultured in a control medium, 10, 5, or 2.5 μM T16A-containing medium for 24 or 48 h. After culture, MTT assays were done using an MTT Cell Proliferation Assay Kit (Cayman). The absorbance of the formazan was measured with a microplate reader (Multiskan Spectrum, Thermo Fisher) at 570 nm.

**Chloride imaging**. NHEKs were seeded on glass-bottom dishes (Matsunami). Cells were incubated with 10 mM MQAE (chloride ion-quenched fluorescent indicator, Dojindo) for 60 min at 37 °C in a Stage Top Incubator (TOKAI HIT) on a confocal laser scanning microscope (LSM 510META, Carl Zeiss). MQAE was excited at 780 nm using a two-photon excitation laser system (Mai Tai, Spectra-Physics), and emission was at 458–479 nm.

The 0 mM chloride calibration solution contained 10 mM NaNO₃, 140 mM KNO₃, 0.5 mM Ca(NO₃)₂, 0.5 mM Mg(NO₃)₂, 10 mM HEPES, and 5 mM D-glucose at pH 7.4, adjusted with CsOH. The 100 mM chloride calibration solution contained 10 mM NaNO₃, 100 mM KCl, 40 mM KNO₃, 0.5 mM

Ca(NO₃)₂, 0.5 mM Mg(NO₃)₂, 10 mM HEPES, and 5 mM D-glucose at pH 7.4, adjusted with CsOH. The 50 mM chloride calibration solution was made by mixing 0 and 100 mM chloride calibration solution 1:1. Each calibration solution was used by adding nigericin (monovalent cation ionophore), valinomycin (potassium ionophore) and tributyltin (chloride ionophore) so that final concentrations were 5, 10, and 10 μM, respectively. All experiments were done at 37 °C.

**Cell cycle assay**. NHEKs were seeded in the same way as the wound-healing assay. After removing the culture insert at 12 h, cells were stained with a Cell-Clock Cell Cycle Assay Kit (Biocolor). ImageJ software was used for data analysis. The distribution of cell cycle phases was defined as the threshold color. G2/M phase (dark blue) cells were defined as Hlu 0–255, Saturation 40–255, Brightness 0–90. S phase (green) cells were defined as Hlu 70–255, Saturation 40–255, Brightness 90–255. G0/G1 phase (yellow-green) cells were defined as Hlu 0–70, Saturation 40–255, Brightness 90–255.

**Statistics and reproducibility**. Data were expressed as means ± SEM. Statistical analysis was performed using the two-tailed Mann–Whitney test or one-way ANOVA to calculate the significance of differences between the two groups. Bonferroni correction or Dunnett's test was used to calculate the significance of differences between multiple comparisons. $p < 0.05$ was considered to be significant.

**Reporting summary**. Further information on research design is available in the Nature Portfolio Reporting Summary linked to this article.

## Data availability

All data and materials used in the analysis are available in the main text with Figures and Supplementary Information. All uncropped gel images are available in Suppl. Figs. 11. Statistical analysis data were available in Supplementary Data 1. Other data or information that supports the findings of this study are available from the corresponding author M.T. (tominaga@nips.ac.jp), upon request.

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

## Acknowledgements
This work was supported by grants to M.T. from a Grant-in-Aid for Scientific Research from the Ministry of Education, Culture, Sports, Science and Technology in Japan (#21H02667 and #20H05768; Scientific Research on Innovative Areas "Thermal Biology").

## Author contributions
Y.Y., Y.T., J.L., and M.T. designed experiments and interpreted data. Y.Y. and J.L. performed experiments and analyzed the data. S.H. and Y.M. contributed to the chloride imaging experiment. The manuscript was prepared by Y.Y., Y.T., and M.T. All authors discussed the results and commented on the manuscript.

## Competing interests
The authors declare no competing interests.
