## [Peer Review File · Communications Biology]

Reviewers' comments:

Reviewer #1 (Remarks to the Author):

The current manuscript by Yamanoi et al., reported the interaction of TRPV3 and ANO1 and the inhibitory effect of inhibition of ANO1 channels on cell migration and proliferation in NHEK cells. The finding is interesting. However, many issues need to be addressed,

1: TRPV3 has been shown to be trigger cell proliferation as well as cell death in keratinocytes (PMID: 32535744). Moderate activation of TRPV3 triggers proliferation while robust activation of TRPV3 induces cell death as seen by G573S mutant. This is the most relevant finding to this manuscript, but the author didn't mention this publication.

2: The author used very high concentration of camphor to induce the currents in HEK-293 cells and nicely proof that TRPV3 activation leads to ANO1 activation. However, in NHEK cells, things may be much more complicated. There are no control experiments showing that camphor-induced currents are dependent on TRPV3 (maybe some, but not exclusively). In Fig. 3F, ANO1 inhibitor only suppresses 20%-30% camphor-induced Cl⁻ currents. Thus, the question is that what is the contribution of TRPV3/ANO1 signaling in the contribution of cell proliferation. The author should do more experiments to address this.

3: In Figure 6C, I am wondering if you can see a Cl⁻ concentration increase after exposed to camphor or ANO1 activator and the suppression of this increase by TRPV3 or ANO1 inhibitor or RNA interference.

4: From the Figure 5, it appears that the ANO1 channel is endogenously active. The question is that does TRPV3 inhibitor itself inhibit ANO1 activity without camphor activation. Or does that only happen during camphor activation. It should be noted, in the culture system, since the temperature is 37 C, TRPV3 should be active.

5: In fig. 1E, why the TRPC6 currents are so different between HEK-293 cells expressing TRPC6 and NHEK cells. It is better to use an inhibitor to demonstrate that these currents are from TRPC6 channels.

6: In Fig. 2A, 3A-B&F. please explain why after camphor addition, there is a gradually increased inward current in addition to those spikes.

7: The Figure 1 needs to be better illustrated. Some data can be moved to supplemental figures.

8: Please discuss the potential application of this finding.

9: It is better to provide an in vivo study to show ANO1 inhibitor can be used in a proliferative skin disease model.

Reviewer #2 (Remarks to the Author):

The manuscript „TRPV3-ANO1 interaction positively regulates 1 wound healing in keratinocytes“ by Yamanoi et al. is a very well-done work focused on the interaction between TRPV3 and ANO1 channels in keratinocytes. Using molecular biology, electrophysiology, calcium and chloride imaging and microscopy, the authors demonstrate that activation of TRPV3 induces chloride influx through ANO1 and that this interaction could positively regulate keratinocyte proliferation and migration. The insight into the mechanism of the TRPV3-ANO1 functional interaction is novel and could contribute to further elucidate the role of chloride in cell signaling pathways in keratinocytes. The study is well-written and carefully executed, with the typical high standards of the Tominaga group.

I have only four minor suggestions that might help the authors improve the manuscript:

1) Page 8, line 264, please correct NHEKs instead of NHKEs

2) Page 13, Figure 3, panel C. Please swap the axis captions, i.e. x-axis mV, y-axis nA

3) Page 22, Figure 8, panel C. Please change the ratio of the axes in the graph so that the points are not oval. Please delete the space in the legend (10 μ M instead of 10 μ M).

4) Throughout the text there are double spaces after sentences instead of one. Please correct.

Reviewer #3 (Remarks to the Author):

The MS by Yamanori et al describes a calcium-activated chloride ANO1 channel current induced by

TRPV3 channel activation in normal human epidermal keratinocytes (NHEKs) for wound healing. In the paper, authors seemed to hypothesize that TRPV3 and ANO1 channels may interact based on literatures that ANO1 is activated by calcium influx induced by some other TRP channels and also TRPV3 is involved in wound healing and ANO1 playing a role in cell migration of epithelial cancer cells. Author started testing expressions of TRPs and ANOs in NHEKs and found TRPV3 and ANO1 expressed in the cells using RT-PCR and Western blot assays before cell imaging of intracellular calcium rise. They further performed whole-cell patch clamp recordings of HEK293 cell co-expressing TRPV3 and ANO1 and showed TRPV3 agonist camphor-induced currents in TRPV3+ANO1 expressed cells, but weak currents in either TRPV3 or ANO1 alone cells. In NHEKs, they showed the camphor-induced chloride currents that were only partially by ANO1 specific blocker Ani9 at 3 μ M. They used a different ANO1 blocker T16Ainh-A01 (5 μ M) that inhibited cell migration in cell culture insert assay and also induced AMP kinase phosphorylation. Overall, their conclusions for interactions between TRPV3 and ANO1 are not well supported by the presented data in which some critical experiments are missing. Here are some specific comments and suggestions:

1. In Fig 1B, there are two bands and what is the band near at 250 KDa?
2. It is known that hTRPV3 C169S mutant is not responsive to camphor. As control, authors need to test any effect of camphor on inducible currents in HEK293T cells co-expressing hTRPV3 C169S mutant and ANO1.
3. Did authors observe any dose-dependent effect of camphor on induced currents? Any TRPV3 inhibitor was used to confirm the observation? In addition, have authors tested other TRPV3 agonists, such as carvacrol-induced currents?
4. In NHEKs silencing TRPV3, is camphor still able to induce any currents?
5. In Fig 3, are camphor induced currents dose-dependently blocked by Ani9?
6. In fig 2C, is there a labeling error for V3?

Reviewer #4 (Remarks to the Author):

General Comments

This manuscript describes the interaction between TRPV3 and ANO1 channels in normal human skin keratinocytes. After examining the expression profile of several TRP and Anoctamin channels in keratinocytes, the authors show that the TRPV3 agonist Camphor led to a robust elevation of intracellular Ca^{2+} concentration as detected with Fura-2, and activation of Ani-9-sensitive Cl^- currents that are presumably the product of ANO1. Since the pipette solution in whole-cell patch clamp experiments contained a high concentration of the Ca^{2+} chelating agent BAPTA (5 mM), the authors concluded that the two ion channels are co-localized within a narrow membrane microdomain to allow for a significant interaction to occur. Co-expression of TRPV3 and ANO1 in HEK-293 cells showed that either protein could be immunoprecipitated with an antibody specific to the other channel protein. Following these biophysical experiments, the authors then present results examining the role of ANO1 in cell migration and proliferation. They show that the ANO1 blocker T16Ainh-A01 or cell exposure to a low external Cl^- buffer significantly attenuated the migration and proliferation of skin keratinocytes, effects mediated by p38 phosphorylation and cell cycle arrest. Separate experiments with the Cl^- fluorescent indicator MQAE revealed that the intracellular concentration of Cl^- is low in keratinocytes (~ 7 mM), dictating an equilibrium potential for Cl^- at ~ -76 mV, which would drive an influx of Cl^- at the resting membrane potential (RMP) in these cells (-24 to -40 mV). From these experiments, the authors conclude that ANO1 plays an important role in wound healing.

This is a generally well-written manuscript presenting novel and important data, which add to a growing number of studies revealing the existence in various cell types of a compartmentalized interaction between ANO1 and other Ca^{2+} -permeable ion channels such as TRPV1, TRPV4, TRPC1, TRPC6, IP3R and ryanodine receptors. The experiments were well-executed and the results presented are for the most part compelling. There are some concerns with the study that will need to be carefully addressed and which will require additional experiments. The authors exclusively relied on pharmacological agents to assess the role of various TRP channels expressed in

keratinocytes. Given the high concentration (10 mM) of the TRPV3 channel agonist Camphor that was used to determine the role of this channel in activating ANO1 and the relatively large number of TRP and Anoctamin channel subunits expressed in these cells, definitive support for the claim that TRPV3 being the only channel activating ANO1 should include silencing RNA experiments. Another aspect that is concerning is the fact that the first series of experiments dealt with the interaction of TRPV3 and ANO1 (Figs. 1-3), but then the manuscript shifted to focus only the role of ANO1 in cell migration and proliferation (Figs. 4, 5, 7 and 8). This latter part is only incrementally novel as the role of ANO1 in these functions has been well-established in many cell types including cancer cells. What is the role of TRPV3 in activating ANO1 and how is this influencing migration and proliferation? Finally, it is unclear why the authors opted to use another less specific blocker of ANO1 (T16AInh-A01) for these latter experiments.

Specific Comments

a) Major:

1. Figure 1 shows the expression of TRP and Anoctamin channels and the Ca²⁺ response of keratinocytes to various TRP agonists. The main conclusion from these experiments is that Ca²⁺ entry is mainly occurring through TRPV3 and TRPV4 on the basis of cells responding to Camphor and GSK. This is not sufficient to unequivocally conclude that TRPV3 is the main Ca²⁺ entry pathway that is responsible for activation of ANO1 through a local mechanism. First, several specific TRPV4 antagonist exists (e.g., HC 067047). The effects of Camphor in the presence of a TRPV4 antagonist should be carried out and compared to results obtained in its absence. siRNA experiments targeting TRPV3 and TRPV4 should also be performed to corroborate the above results. This is critical because the whole study revolves around the presumably specific effect of Camphor when used at 10 mM at stimulating TRPV3 but not TRPV4 or others, and importantly the fact that a local interaction between ANO1 and TRPV4 was already demonstrated in other cell types by the submitting authors (Y. Takayama, K. Shibasaki, Y. Suzuki, A. Yamanaka, M. Tominaga, Modulation of water efflux through functional interaction between TRPV4 and TMEM16A/anoctamin 1. *FASEB J* 28, 2238-2248 (2014). S. Derouiche, Y. Takayama, M. Murakami, M. Tominaga, TRPV4 heats up ANO1-dependent exocrine gland fluid secretion. *FASEB J* 32, fj201700954R (2018). One series of experiments that is surprisingly missing in view of the scope of this article concerns the effects of Ani9 on the Ca²⁺ response to Camphor. Blocking ANO1 should reduce the magnitude of the Ca²⁺ transient elicited by Camphor since ECl is more negative than RMP in these cells (Lines 465-470 and Figure 6). This would add credence to the concept of a local interaction between TRPV3 and ANO1 and its impact on signaling. Finally, it would be nice if the authors could add a mean bar graph summarizing these results.

2. Figures 2 and 3 present very elegant and convincing data highlighting the interaction between TRPV3 and ANO1. One aspect that is lacking from these figures is that it is not possible to visualize the gating properties of the ANO1 current activated by TRPV3 stimulation. At [Ca²⁺]_i in the submicromolar range, ANO1 currents display slow activation kinetics at positive step potentials and slow deactivation kinetics upon return to the holding potential, and outwardly rectifying properties. These are signature biophysical characteristics that would add strength to the interpretation that the currents elicited are indeed produced by classical CaCCs encoded by ANO1.

3. As alluded to in the General Comments section, it is unclear why the authors decided to discard investigating the role of TRPV3 and its interaction with ANO1 on cell migration and proliferation. The authors will have to repeat similar experiments (Figs. 4, 5, 7 and 8) in the presence of Camphor, and Camphor + Ani9. Without these experiments, the study looks premature and certainly not aligned with the proposed title of the paper. We already know from work done in other cell types that ANO1 is a major contributor to cell migration and proliferation and this has relevance to our understanding of pathophysiological mechanisms in various forms of cancer. Another major concern was the unexplained switch from using the very specific and potent Ani9 to the slightly less potent and now recognized non-specific inhibitor T16AInh-A01. This needs to be justified in the manuscript. Does Ani9 produce similar effects? 1 μM T16AInh-A01 abolished L-type Ca²⁺ current in vascular myocytes (D. M. Boedtkjer, S. Kim, A. B. Jensen, V. M. Matchkov, K. E. Andersson, New selective inhibitors of calcium-activated chloride channels – T16AInh-A01, CaCCInh-A01, and MONNA – What do they inhibit? *Br J Pharmacol* 172, 4158-4172 (2015)). The latter point is important because keratinocytes are known to express voltage-gated Ca²⁺ channels (M. Denda, S. Fujiwara, T. Hibino, Expression of voltage-gated calcium channel subunit alpha1C in

epidermal keratinocytes and effects of agonist and antagonists of the channel on skin barrier homeostasis. *Exp Dermatol* 15, 455-460 (2006)), which begs the question about their potential interaction with ANO1, and role in cell migration and proliferation. Are these functional effects altered by dihydropyridines?

4. Figure 6: Although I understand why the authors performed these experiments, as presented they do not add much to the story as presented because the results are static. They could be improved by showing how blocking ANO1 on its own affects $[Cl]_i$ and how stimulation of TRPV3 with Camphor affects this measurement and whether blocking ANO1 in the presence of Camphor alters the response. The manuscript would be strengthened by adding an original trace displaying the effects of drug applications on $[Cl]_i$. This goes back to my earlier comment about refocusing the later part of the study on the interaction of TRPV3 and ANO1.

b) Minor:

1. Line 64: "lachrymal"; should read: "lacrimal".

2. Line 85: "... function of ANO1 in skin normal keratinocyte ..."; this sounds awkward. I suggest an inversion: "... function of ANO1 in NORMAL SKIN keratinocyte ...".

3. Lines 97-98: "The following antibodies were used: rabbit anti-ANO1 antibody (Abcam, ab53213, 1:5), (Abcam, ab53212, 1:00), ..."; are these antibody dilutions correct? 1:5 seems very awfully high and I don't know what 1:00 means. Please verify.

4. Lines 247-248: "... Student's t-test for calculate differences between two groups. Bonferroni correction or Dunnett's test was used for calculate differences ..."; more appropriately: "... Student's t-test TO calculate differences between two groups. Bonferroni correction or Dunnett's test was used TO calculate differences ..."

5. Line 347: "... positive direction when the extracellular chloride ..."; better wording: "positive POTENTIAL when the extracellular chloride ...".

6. Line 350: "... might not be a major contributor to ..."; suggestion for improvement: "... WAS NOT a major contributor to ...".

7. Line 409: "... chloride concentrations should be reduced upon depletion of extracellular chlorides23."; more appropriately: "... chloride CONCENTRATION should be reduced upon depletion of extracellular CHLORIDE23."

8. Figure 4A: The legend for the culture insert is shown as a white box. Shouldn't it be a light blue box for clarity's sake?

9. Line 456: "... intracellular chloride concentrations and membrane potentials."; more appropriately: "... intracellular chloride CONCENTRATION and membrane POTENTIAL."

10. Line 465: "... performed a chloride-imaging experiment using a chloride indicator, MQAE25-27."; suggestion for improvement: "... performed CHLORIDE-IMAGING EXPERIMENTS using THE chloride indicator, MQAE25-27."

11. Line 469: "... that chloride influx occurred through ANO1 in NHEKs at the reported resting ..."; awkward sentence. Suggestion: "... that chloride influx through ANO1 WOULD OCCUR AT THE REPORTED RESTING MEMBRANE POTENTIAL IN NHEKs (-24 to -40 mV)28-30."

12. Figure 8: I am unsure how to visualize the pictures shown in panel B. I was initially reading them in the opposite direction. I think it would make more sense and be less counterintuitive to report the % for the actual cycle phase that is targeted rather than the red cells.

13. Lines 580-581: "CaCCInh-A01 could utilize a unique mechanism for inhibiting cell proliferation without affecting ANO1 channel activity." This sentence seems to be coming out of nowhere and out of context. I could not follow the logic with this ANO1 inhibitor.

14. Since the authors did not take advantage of advanced microscopy techniques (e.g., super-resolution nanomicroscopy, d-STORM, STED and others) and solely relied on Co-IP and patch clamp experiments to support the claim of a co-localization of TRPV3 and ANO1 channels, it would be useful and warranted to add a short paragraph in the Discussion speculating on the distance separating the two channels of interest based on models of Ca^{2+} diffusion from a single point source that includes fixed and mobile (in this case BAPTA) buffers (e.g., G. D. Smith, J. Wagner, J. Keizer, Validity of the rapid buffering approximation near a point source of calcium ions. *Biophys J* 70, 2527-2539 (1996)).

Reviewer #1 (Remarks to the Author):

1: TRPV3 has been shown to be trigger cell proliferation as well as cell death in keratinocytes (PMID: 32535744). Moderate activation of TRPV3 triggers proliferation while robust activation of TRPV3 induces cell death as seen by G573S mutant. This is the most relevant finding to this manuscript, but the author didn't mention this publication.

Response:

We thank the reviewer for this comment. We have added the work and discussed it in the revised manuscript.

2: The author used very high concentration of camphor to induce the currents in HEK-293 cells and nicely proof that TRPV3 activation leads to ANO1 activation. However, in NHEK cells, things may be much more complicated. There are no control experiments showing that camphor-induced currents are dependent on TRPV3 (maybe some, but not exclusively). In Fig. 3F, ANO1 inhibitor only suppresses 20%-30% camphor-induced Cl⁻ currents. Thus, the question is that what is the contribution of TRPV3/ANO1 signaling in the contribution of cell proliferation. The author should do more experiments to address this.

Response:

In accord with the reviewer's suggestion, we performed patch-clamp experiments and examined the effects of a new TRPV3 inhibitor and a high concentration (30 μM) of Ani9, another ANO1 inhibitor, on camphor-induced currents in NHEK cells. A TRPV3 inhibitor, dyclonine, suppressed the camphor-induced currents, albeit at high concentrations, suggesting TRPV3-mediated responses. At a high concentration, the ANO1 inhibitor, Ani9, also almost completely suppressed the camphor-induced currents. These data strongly suggest that the TRPV3/ANO1 complex functions in NHEK cells, as well. Accordingly, we have added the data with dyclonine (Supplementary Figure 3) and replaced the previous data (using a low concentration (3 μM) of Ani9) with new data (Figure 3F) in the revised manuscript.

3: In Figure 6C, I am wondering if you can see a Cl⁻ concentration increase after exposed to camphor or ANO1 activator and the suppression of this increase by TRPV3 or ANO1 inhibitor or RNA interference.

Response:

We thank the reviewer for the suggestion. Long-term exposure to camphor caused cell volume changes in chloride imaging, which led to changes in fluorescence intensity. As a result, we could not accurately evaluate the changes in cytosolic Cl⁻ concentrations. In addition, we suspect that Cl⁻ influx caused by ANO1 is not sufficient to cause global intracellular Cl⁻ concentration changes, and that changes in Cl⁻ concentrations in the vicinity of the plasma membrane may be important.

4: From the Figure 5, it appears that the ANO1 channel is endogenously active. The question is that does TRPV3 inhibitor itself inhibit ANO1 activity without camphor activation. Or does that only happen during camphor activation. It should be noted, in the culture system, since the temperature is 37 C, TRPV3 should be active.

Response:

We thank the reviewer for raising these points. The data shown in the original Figure 5 indicates that TRPV3 is active even without external chemical activators. In order to prove it, we performed culture insert assays with a TRPV3 inhibitor, dyclonine in the absence of any TRPV3 activator. Dyclonine inhibited cell migration/proliferation, indicating that TRPV3 is involved in cell migration/proliferation in this assay. It should be noted that we used newly thawed NHEK cells in the experiment for the revision. Therefore, the migration and/or proliferation of the cells were higher than before, and the control group showed different values compared with the data in the original manuscript. We have added the data with dyclonine to the revised Supplementary Figure 4.

5: In fig. 1E, why the TRPC6 currents are so different between HEK-293 cells expressing TRPC6 and NHEK cells. It is better to use an inhibitor to demonstrate that these currents are from TRPC6 channels.

Response:

We appreciate this question. The reviewer might refer to TRPV6. Although we detected TRPV6 mRNA in NHEK cells, we do not think it is functionally expressed. If TRPV6 is functionally expressed in NHEK cells, intracellular calcium concentrations should decrease in the calcium (-) solution; this was not observed in our experiment (Fig. 1D). In addition, TRPV6-mediated currents are known to exhibit an inward rectification as shown in Figure 1E. However, such currents were not observed in NHEK cells. These data suggest TRPV6 is not functionally expressed in NHEK cells. We have clarified this point in the revised manuscript.

6: In Fig. 2A, 3A-B&F. please explain why after camphor addition, there is a gradually increased inward current in addition to those spikes.

Response:

In this experiment, we did not examine TRPV3-mediated currents. Instead, ANO1-mediated chloride currents were observed because we used NMDG-Cl solution for both bath and pipette solutions. There was a gradual increase in intracellular calcium entering the cells, probably through TRPV3 that activates ANO1. In addition, TRPV3-mediated currents are known to show

gradual increases. This could explain the gradual increase in chloride currents in Figures 2A, 3A-B&F.

7: The Figure 1 needs to be better illustrated. Some data can be moved to supplemental figures.

Response:

In accord with the reviewer's suggestion, we have moved negative data shown in the original Figure 1C to Supplementary Figure 2.

8: Please discuss the potential application of this finding.

Response:

We thank the reviewer for the important suggestion. We have added some discussion in the revised manuscript.

9: It is better to provide an in vivo study to show ANO1 inhibitor can be used in a proliferative skin disease model.

Response:

We agree that it is better to examine the involvement of TRPV3-ANO1 interaction *in vivo*. However, all experiments in this manuscript were performed with hTRPV3 and human keratinocytes. We would like to examine the mouse model in the future.

Reviewer #2 (Remarks to the Author):

- 1) Page 8, line 264, please correct NHEKs instead of NHKEs
- 2) Page 13, Figure 3, panel C. Please swap the axis captions, i.e. x-axis mV, y-axis nA
- 3) Page 22, Figure 8, panel C. Please change the ratio of the axes in the graph so that the points are not oval. Please delete the space in the legend (10 μ M instead of 10 μ M).
- 4) Throughout the text there are double spaces after sentences instead of one. Please correct.

Response:

We thank the reviewer for pointing out those errors. We have fixed all of them in the revised manuscript.

Reviewer #3 (Remarks to the Author):

1. In Fig 1B, there are two bands and what is the band near at 250 KDa?

Response:

We do not know exactly what it is. The upper band could be a dimer because it is at nearly twice the molecular weight of monomer ANO1 protein.

2. It is known that hTRPV3 C169S mutant is not responsive to camphor. As control, authors need to test any effect of camphor on inducible currents in HEK293T cells co-expressing hTRPV3 C169S mutant and ANO1.

Response:

As the reviewer pointed out, the experiment with the hTRPV3 C169S mutant effectively proved the involvement of TRPV3. However, camphor-induced currents were not observed in HEK293T cells not expressing hTRPV3, strongly suggesting that the observed chloride currents are activated downstream of hTRPV3 activation. Therefore, we do not think that further control experiments are needed.

3. Did authors observe any dose-dependent effect of camphor on induced currents? Any TRPV3 inhibitor was used to confirm the observation? In addition, have authors tested other TRPV3 agonists, such as carvacrol-induced currents?

Response:

We observed chloride currents with 10 mM camphor in NHEK cells, but not with 3 or 6 mM, as shown below. The data suggest that strong activation of TRPV3 by 10 mM camphor (that is, a large increase in intracellular calcium concentrations by TRPV3) is necessary to activate ANO1.

We examined the effect of a TRPV3 inhibitor on camphor-induced currents in NHEK cells as suggested. A TRPV3 inhibitor, dyclonine, suppressed the camphor-induced currents, albeit at high concentrations, suggesting that this current depends on TRPV3. The data have been placed in Supplementary Figure 3. We previously reported that carvacrol inhibits ANO1¹. We also

confirmed that carvacrol inhibits ANO1-mediated currents in NHEK cells as shown below. Therefore, we did not perform the experiment.

1. Takayama, Y., Furue, H. & Tominaga, M. 4-isopropylcyclohexanol has potential analgesic effects through the inhibition of anoctamin 1, TRPV1 and TRPA1 channel activities. *Sci Rep* 7, 43132, doi:10.1038/srep43132 (2017).

4. In NHEKs silencing TRPV3, is camphor still able to induce any currents?

Response:

Silencing experiments were performed in which we targeted ANO1, TRPV3 and TRPV4 in NHEK cells as shown below. However, we did not observe a reduction of ANO1 at the protein level in Western blotting (see below), even though siRNA (1162) strongly reduced ANO1 mRNA (below). In terms of siRNA treatment of TRPV3, we failed to find an effective siRNA that would reduce the mRNA level. Instead of the silencing experiments, we performed patch-clamp experiments with a TRPV3 inhibitor focusing on camphor-induced currents in NHEK cells. A TRPV3 inhibitor, dyclonine, suppressed camphor-induced currents, albeit at high concentrations, which should be sufficient to prove the involvement of TRPV3. The data have been added in the revised Supplementary Figure 3.

RT-PCR of NHEK cells in knock down condition

Western blotting in knock down condition

5. In Fig 3, are camphor induced currents dose-dependently blocked by Ani9?

Response:

We thank the reviewer for the input. As suggested, we examined the effects of a high concentration of Ani9 on camphor-induced currents in NHEK cells. At 30 μ M, Ani9 almost completely suppressed the camphor-induced currents. Accordingly, we have replaced the previous data utilizing a low concentration (3 μ M) of Ani9 with the new data (Figure 3F) in the revised manuscript.

6. In fig 2C, is there a labeling error for V3?

Response:

We thank the reviewer for pointing this out. We have corrected the labels.

Reviewer #4 (Remarks to the Author):

1. Figure 1 shows the expression of TRP and Anoctamin channels and the Ca²⁺ response of keratinocytes to various TRP agonists. The main conclusion from these experiments is that Ca²⁺ entry is mainly occurring through TRPV3 and TRPV4 on the basis of cells responding to Camphor and GSK. This is not sufficient to unequivocally conclude that TRPV3 is the main Ca²⁺ entry pathway that is responsible for activation of ANO1 through a local mechanism. First, several specific TRPV4 antagonist exists (e.g., HC 067047). The effects of Camphor in the presence of a TRPV4 antagonist should be carried out and compared to results obtained in its absence. siRNA experiments targeting TRPV3 and TRPV4 should also be performed to corroborate the above results. This is critical because the whole study revolves around the presumably specific effect of Camphor when used at 10 mM at stimulating TRPV3 but not TRPV4 or others, and importantly the fact that a local interaction between ANO1 and TRPV4 was already demonstrated in other cell types by the submitting authors (Y. Takayama, K. Shibasaki, Y. Suzuki, A. Yamanaka, M. Tominaga, Modulation of water efflux through functional interaction between TRPV4 and TMEM16A/anoctamin 1. *FASEB J* 28, 2238-2248 (2014). S. Derouiche, Y. Takayama, M. Murakami, M. Tominaga, TRPV4 heats up ANO1-dependent exocrine gland fluid secretion. *FASEB J* 32, fj201700954R (2018).

One series of experiments that is surprisingly missing in view of the scope of this article concerns the effects of Ani9 on the Ca²⁺ response to Camphor. Blocking ANO1 should reduce the magnitude of the Ca²⁺ transient elicited by Camphor since ECl is more negative than RMP in these cells (Lines 465-470 and Figure 6). This would add credence to the concept of a local interaction between TRPV3 and ANO1 and its impact on signaling. Finally, it would be nice if

the authors could add a mean bar graph summarizing these results.

Response:

Silencing experiments were performed in which we targeted ANO1, TRPV3 and TRPV4 in NHEK cells. We did not observe a reduction of ANO1 protein in Western blotting even under conditions in which the siRNA (1162) strongly reduced *ANO1* mRNA (below). In terms of siRNA targeting of TRPV3, we failed to find a good siRNA species that could reduce the mRNA level. Instead of the silencing experiments, we performed patch-clamp experiments with a TRPV3 inhibitor focusing on camphor-induced currents in NHEK cells. A TRPV3 inhibitor, dyclonine, suppressed camphor-induced currents, albeit at high concentrations, which could be sufficient to prove the involvement of TRPV3. Because the involvement of TRPV3 was clearly proven by the experiment with dyclonine, we decided not to do the experiment with a TRPV4 antagonist. The data have been added in the revised Supplementary Figure 3.

We undertook calcium imaging of HEK293T cells with camphor in the presence of ANO1 inhibitors, T16A or Ani9. The calcium responses with 6 mM of camphor were not changed very much in the presence of ANO1 inhibitors. The changes in intracellular chloride concentrations due to ANO1 inhibition were insufficient to significantly alter calcium influx. Therefore, we believe that chloride influx through ANO1 exerts its physiological function through changes in local chloride concentrations near the plasma membrane.

2. Figures 2 and 3 present very elegant and convincing data highlighting the interaction between TRPV3 and ANO1. One aspect that is lacking from these figures is that it is not possible to visualize the gating properties of the ANO1 current activated by TRPV3 stimulation. At $[Ca^{2+}]_i$ in the submicromolar range, ANO1 currents display slow activation kinetics at positive step potentials and slow deactivation kinetics upon return to the holding potential, and outwardly rectifying properties. These are signature biophysical characteristics that would add strength to the interpretation that the currents elicited are indeed produced by classical CaCCs encoded by ANO1.

Response:

In accord with the reviewer's suggestion, we analyzed the properties of whole-cell currents activated by high intracellular free calcium (500 nM) in NHEK cells by applying step pulses. The currents showed slow activation kinetics, characteristic of ANO1 as pointed out by the reviewer. We have added the result in the revised Supplementary Figure 1.

3. As alluded to in the General Comments section, it is unclear why the authors decided to discard investigating the role of TRPV3 and its interaction with ANO1 on cell migration and proliferation. The authors will have to repeat similar experiments (Figs. 4, 5, 7 and 8) in the presence of Camphor, and Camphor + Ani9. Without these experiments, the study looks premature and certainly not aligned with the proposed title of the paper. We already know from work done in other cell types that ANO1 is a major contributor to cell migration and proliferation and this has relevance to our understanding of pathophysiological mechanisms in various forms of cancer. Another major concern was the unexplained switch from using the very specific and potent Ani9 to the slightly less potent and now recognized non-specific inhibitor T16AInh-A01. This needs to be justified in the manuscript. Does Ani9 produce similar effects? 1 μ M T16AInh-A01 abolished L-type Ca^{2+} current in vascular myocytes (D. M. Boedtkjer, S. Kim, A. B. Jensen, V. M. Matchkov, K. E. Andersson, New selective inhibitors of calcium-activated chloride channels – T16AInh-A01, CaCCInh-A01, and MONNA – What do they inhibit? *Br J Pharmacol* 172, 4158-4172 (2015)). The latter point is important because keratinocytes are known to express voltage-gated Ca^{2+} channels (M. Denda, S. Fujiwara, T. Hibino, Expression of voltage-gated calcium channel subunit α_1C in epidermal keratinocytes and effects of agonist and antagonists of the channel on skin barrier homeostasis. *Exp Dermatol* 15, 455-460 (2006)), which begs the question about their potential interaction with ANO1, and role in cell migration and proliferation. Are these functional effects altered by dihydropyridines?

Response:

As the reviewer suggested, we performed culture insert assays with camphor or Ani9. Unfortunately, camphor showed toxic effects during the long cultivation necessary for the culture

insert assay. On the other hand, Ani9 inhibited cell migration/proliferation, suggesting the involvement of ANO1. In order to strengthen the data for the involvement of TRPV3, we used a TRPV3 inhibitor (dyclonine) in the culture insert assay. Dyclonine inhibited cell migration/proliferation. These data strongly suggest the involvement of both TRPV3 and ANO1 in migration/proliferation of NHEK cells. It should be noted that we used newly thawed NHEK cells in the experiments for the revised paper. Therefore, the migration and/or proliferation of the cells were higher than before, and the control group showed different values compared with the data in the previous manuscript. We have added those data to the revised Supplementary Figures 4 and 6.

Concerning the involvement of voltage-gated Ca^{2+} channels, we performed patch-clamp experiments by depolarizing the membrane potentials or calcium-imaging experiments with a high K^+ solution. We did not observe the current responses or increases in cytosolic Ca^{2+} concentrations, respectively, as shown below, indicating that voltage-gated Ca^{2+} channels are not functionally expressed in NHEK cells.

(A) Representative traces of NHEK cells. Step pulses for 500 msec were applied between -100 mV and +100 mV with 20 mV increments from -60 mV. NMDG-Cl base bath and pipette solutions were used. Barium was used instead of Calcium to avoid ANO1 activation.

(B) Calcium imaging of NHEKs cells upon 80 mM potassium bath solution (High K^+).

4. Figure 6: Although I understand why the authors performed these experiments, as presented they do not add much to the story as presented because the results are static. They could be improved by showing how blocking ANO1 on its own affects $[\text{Cl}]_i$ and how stimulation of TRPV3 with Camphor affects this measurement and whether blocking ANO1 in the presence of Camphor alters the response. The manuscript would be strengthened by adding an original trace

displaying the effects of drug applications on $[Cl^-]_i$. This goes back to my earlier comment about refocusing the later part of the study on the interaction of TRPV3 and ANO1.

Response:

We thank the reviewer for the suggestion. Long-term exposure to camphor caused cell volume changes in chloride imaging, which led to changes in fluorescence intensity. Thus, we could not accurately evaluate the changes in cytosolic Cl^- concentrations. In addition, we suspect that Cl^- influx by ANO1 is not sufficient to cause global intracellular Cl^- concentration changes, and that changes in Cl^- concentrations in the vicinity of the plasma membrane may be important.

Minor:

1. Line 64: “lachrymal”; should read: “lacrimal”.

Response:

We thank the reviewer for pointing this out. We have corrected it.

2. Line 85: “... function of ANO1 in skin normal keratinocyte ...”; this sounds awkward. I suggest an inversion: “... function of ANO1 in NORMAL SKIN keratinocyte ...”.

Response:

We thank the reviewer for bringing this to our attention. We have corrected it.

3. Lines 97-98: “The following antibodies were used: rabbit anti-ANO1 antibody (Abcam, ab53213, 1:5), (Abcam, ab53212, 1:00)”; are these antibody dilutions correct? 1:5 seems very awfully high and I don’t know what 1:00 means. Please verify.

Response:

We used “ab53213” at a high concentration and it is a pre-diluted antibody. “1:00” was an error, and it should be 1:100. It has been corrected.

4. Lines 247-248: “... Student’s t-test for calculate differences between two groups. Bonferroni correction or Dunnett’s test was used for calculate differences ...”; more appropriately: “... Student’s t-test TO calculate differences between two groups. Bonferroni correction or Dunnett’s test was used TO calculate differences ...”

Response:

We thank the reviewer for pointing this out. We have corrected it.

5. Line 347: “... positive direction when the extracellular chloride ...”; better wording: “positive POTENTIAL when the extracellular chloride ...”.

Response:

We thank the reviewer for pointing this out. We have corrected it.

6. Line 350: "... might not be a major contributor to ..."; suggestion for improvement: "... WAS NOT a major contributor to ...".

Response:

We thank the reviewer for suggesting this improvement. We have corrected it.

7. Line 409: "... chloride concentrations should be reduced upon depletion of extracellular chlorides²³."; more appropriately: "... chloride CONCENTRATION should be reduced upon depletion of extracellular CHLORIDE²³."

Response:

We thank the reviewer for pointing this out. We have corrected it.

8. Figure 4A: The legend for the culture insert is shown as a white box. Shouldn't it be a light blue box for clarity's sake?

Response:

We thank the reviewer for bringing this to our attention. We have corrected it.

9. Line 456: "... intracellular chloride concentrations and membrane potentials."; more appropriately: "... intracellular chloride CONCENTRATION and membrane POTENTIAL."

Response:

We thank the reviewer for bringing this issue to our attention. We have corrected it.

10. Line 465: "... performed a chloride-imaging experiment using a chloride indicator, MQAE25-27."; suggestion for improvement: "... performed CHLORIDE-IMAGING EXPERIMENTS using THE chloride indicator, MQAE25-27."

Response:

We thank the reviewer for pointing this out. We have corrected it.

11. Line 469: "... that chloride influx occurred through ANO1 in NHEKs at the reported resting ..."; awkward sentence. Suggestion: "... that chloride influx through ANO1 WOULD OCCUR AT THE REPORTED RESTING MEMBRANE POTENTIAL IN NHEKs (-24 to -40 mV)²⁸⁻³⁰."

Response:

We thank the reviewer for pointing this out. We have corrected it.

12. Figure 8: I am unsure how to visualize the pictures shown in panel B. I was initially reading them in the opposite direction. I think it would make more sense and be less counterintuitive to report the % for the actual cycle phase that is targeted rather than the red cells.

Response:

We have added a sentence to the legend to avoid misunderstanding and enlarged the % display of Figure 8.

13. Lines 580-581: “CaCCInh-A01 could utilize a unique mechanism for inhibiting cell proliferation without affecting ANO1 channel activity.” This sentence seems to be coming out of nowhere and out of context. I could not follow the logic with this ANO1 inhibitor.

Response:

We had included the above remarks in the original manuscript because a previous study showed that CaCCInh-A01 induced ANO1 protein degradation. Therefore, we intentionally did not use CaCCInh-A01. However, this point is not terribly important and the description could confuse the readers as the reviewer pointed out. Accordingly, we have deleted the description in the revised manuscript.

14. Since the authors did not take advantage of advanced microscopy techniques (e.g., super-resolution nanomicroscopy, d-STORM, STED and others) and solely relied on Co-IP and patch clamp experiments to support the claim of a co-localization of TRPV3 and ANO1 channels, it would be useful and warranted to add a short paragraph in the Discussion speculating on the distance separating the two channels of interest based on models of Ca^{2+} diffusion from a single point source that includes fixed and mobile (in this case BAPTA) buffers (e.g., G. D. Smith, J. Wagner, J. Keizer, Validity of the rapid buffering approximation near a point source of calcium ions. *Biophys J* 70, 2527-2539 (1996)).

Response:

We thank the reviewer for raising the point. We have added some discussion in the revised manuscript.

Reviewers' comments:

Reviewer #1 (Remarks to the Author):

The revised manuscript from Tominaga group has been improved a lot. However, several critical questions have not been addressed. 1: As raised in the previous comments, how to deal with the activation of TRPV3 and inhibition of TRPV3 was not addressed. In the current manuscript, TRPV3 was activated by 10 mM camphor, it has been well established that camphor has many targets including TRPV1 and M8 and maybe many more. Since there is no selective TRPV3 activator available, to activate TRPV3, researcher typically used a combination of activators at relative low concentrations. The author used Dyclonine as a TRPV3 antagonist, although it was reported to be inhibit TRPV3 with 2-3 μM apparent affinity in the E-life paper (PMID: 33876725), a most recent Nature communications paper indicated the apparent affinity is around 30 μM (PMID: 35589741). As a local anesthetics, I suspect if dyclonine only suppresses TRPV3 at 100 μM as used in the current manuscript. There is a serial of better TRPV3 antagonist (although may not be selective) published in Journal of Medicinal Chemistry (PMID: 27077528) with low nanomolar apparent affinity. It is probably better to use 5a to probe the inhibition of TRPV3. Because the unavailability of TRPV3 agonist or antagonist, to over expression of a gain of functional mutant of TRPV3 or silencing TRPV3 mRNA will be good choices.

2: As mentioned in Figure 6C, as respond by authors, "long term exposure of camphor leads to cell swelling. So no Cl^- trace could be recorded". That again raised the specificity of camphor on TRPV3. Additionally, do we need to wait a long time to record the Cl^- trace using fluorescent dye? if yes, why?

In summary, without clearly proving camphor response (both patch recording and Ca^{2+} imaging) is solely dependent on TRPV3, the conclusion of this manuscript is discounted.

Reviewer #3 (Remarks to the Author):

Most of my concerns have been addressed in this revision except one regarding a transfection of hTRPV3 C169S mutant that is not responsive to agonist camphor and serves as a control. Instead, authors presented the data in which camphor induced currents were inhibited by blocker dyclonine, which is acceptable.

Reviewer #4 (Remarks to the Author):

General Comments

Although the authors have satisfactorily addressed many but not all concerns. Actually, a new one surfaced based on new data provided to other reviewers. Major issues concern the pharmacological approach and the use of very high concentrations of agents to activate or block TRPV3, or block ANO1. The authors also failed to convincingly discard the possibility that TRPV4 might also be a participant in the responses to Camphor.

Specific Comment

1. I appreciate the efforts undertaken by the authors to perform silencing RNA experiments to attempt to knock down TRPV3, TRPV4 and ANO1. It is surprising that not a single RNA molecule tested failed to knock down any of these ion channels at the protein level but this occurs sometimes, which can be frustrating. The authors elected instead to test the effects of the TRPV3 inhibitor dyclonine. However, they used massive concentrations (50-500 μM) to demonstrate an effect. This agent was shown to block TRPV3 currents with an IC_{50} of only 3.2 μM (Liu Q, Wang J, Wei X, Hu J, Ping C, Gao Y, Xie C, Wang P, Cao P, Cao Z, Yu Y, Li D, and Yao J. Therapeutic inhibition of keratinocyte TRPV3 sensory channel by local anesthetic dyclonine. eLife 10: e68128, 2021). This raises then raises the question about specificity. Does 500 μM have any direct effect

on either TRPV4 or ANO1 currents? The diclonine experiments do not prove that TRPV4 is not involved in the responses. I had suggested to use the specific and potent TRPV4 inhibitor HC067047 (1-10 μM), which would be a satisfactory alternative to siRNA experiments. The authors must perform these experiments to unequivocally discard the possibility that the activity of Camphor is associated with activation of TRPV4.

2. In the rebuttal, the authors provided a new figure showing that both T16A and Ani9 failed to alter Ca^{2+} responses elicited by 6 mM Camphor. First and foremost, if these two channels are so close to each other, wouldn't you expect a very large effect considering the large Ca^{2+} responses seen? Second, intracellular Cl^- potentially could change (or not), but the more important relationship between the two is related to the effects of ANO1 on membrane potential. Because ECl is very negative and presumably RMP as well, then blocking ANO1 should depolarize the cells leading to less Ca^{2+} entry through TRPV3 due to a decreased driving force for Ca^{2+} . Third, since 6 mM Camphor produced a large Ca^{2+} response, why wasn't ANO1 activated? In your response to reviewer #3, 3 or 6 mM Camphor had no effect on the current. This makes little pharmacological sense. The difference between 6 and 10 mM on a log scale is very small. Is the higher concentration producing some deleterious non-specific effects (the authors mentioned swelling of the cells), which are then partially blocked by the high concentrations of agents used?

3. In their answer to my 2nd major point about the biophysical properties of ANO1 when triggered by activation of TRPV3 with Camphor, the authors performed a simple Ca^{2+} clamp (500 nM) experiment to show that the currents were time-dependent. I never questioned the functional expression of ANO1 in these cells. All I asked for was for the authors to demonstrate that activation of TRPV3 with Camphor leads to the appearance of time-dependent currents consistent with ANO1. The authors should use a repetitive step protocol, rather than a ramp protocol, to show that time-dependent CaCC currents are activated during exposure to Camphor.

4. Figure 3: The authors used 30 μM Ani9 to block the current induced by Camphor. This concentration is excessively high. Ani9 was shown to block ANO1 currents with an IC_{50} of only ~ 110 nM (Seo Y, Lee HK, Park J, Jeon DK, Jo S, Jo M, and Namkung W. Ani9, A Novel Potent Small-Molecule ANO1 Inhibitor with Negligible Effect on ANO2. PLoS One 11: e0155771, 2016) and investigators in the field generally use 1 μM , which has been shown to abolish currents in most cell types. This questions whether the current activated by Camphor is entirely mediated by TRPV3 and ANO1. Since Camphor elicits swelling, is the conductance also comprising a swelling-activated Cl^- current? Does tamoxifen block the current induced by Camphor?

Reviewer #1 (Remarks to the Author):

The revised manuscript from Tominaga group has been improved a lot. However, several critical questions have not been addressed.

1: As raised in the previous comments, how to deal with the activation of TRPV3 and inhibition of TRPV3 was not addressed. In the current manuscript, TRPV3 was activated by 10 mM camphor, it has been well established that camphor has many targets including TRPV1 and M8 and maybe many more. Since there is no selective TRPV3 activator available, to activate TRPV3, researcher typically used a combination of activators at relative low concentrations. The author used Dyclonine as a TRPV3 antagonist, although it was reported to be inhibit TRPV3 with 2-3 μ M apparent affinity in the E-life paper (PMID: 33876725), a most recent Nature communications paper indicated the apparent affinity is around 30 μ M (PMID: 35589741). As a local anesthetics, I suspect if dyclonine only suppresses TRPV3 at 100 μ M as used in the current manuscript. There is a serial of better TRPV3 antagonist (although may not be selective) published in Journal of Medicinal Chemistry (PMID: 27077528) with low nanomolar apparent affinity. It is probably better to use 5a to probe the inhibition of TRPV3. Because the unavailability of TRPV3 agonist or antagonist, to over expression of a gain of functional mutant of TRPV3 or silencing TRPV3 mRNA will be good choices.

Response: We thank the reviewer for pointing this out, and we understand his concern about the nonspecific effects of camphor. However, as shown in Fig.1 and Suppl. Fig.2, TRPV1 and TPRM8 are not functionally expressed in NHEK. Furthermore, when we asked whether TPRV4 contributed to the camphor-induced currents, we found that 10 mM camphor did not activate hTRPV4 and that HC067047, a selective inhibitor of TRPV4, did not suppress the 10 mM camphor-induced current in NHEK. In addition, HC067047 did not affect cell migration and/or proliferation in the wound healing assay with NHEK. We have added these data as revised Suppl. Figs. 3 and 7. Based on these results, we were able to rule out the possibility that the currents induced by 10 mM camphor and the results in the wound healing assay with NHEK were caused by activation of TRPV4.

We tried to obtain the inhibitors 5a and 74a shown in the paper that the reviewer mentioned. However, we were unable to obtain them from AbbVie Medical Research Support Team. We surprisingly found that dyclonine suppressed the ANO1-mediated currents at 500 μ M, but not at 100 μ M (as shown below).

Experimental conditions:

hANO1 expressing HEK cells

HP: 0 mV (-100 mV to + 100 mV)

Bath solution: NMDG-Cl

Pipette solution: NMDG-Cl (500 nM free Ca^{2+})

The previous Supplementary Figure 3 in which 500 μM dyclonine almost completely inhibited 10 mM camphor-induced currents can be interpreted as a mixture of the inhibition of TRPV3-mediated and ANO1-mediated currents. Therefore, we incubated NHEK with 100 μM dyclonine for 5 minutes and no current activation was observed even in the presence of 10 mM camphor, suggesting that TRPV3 does contribute to the camphor-induced currents. Accordingly, we have replaced the previous Suppl. Fig. 3 with a new one (re-revised Suppl. Fig. 4).

2: As mentioned in Figure 6C, as respond by authors, "long term exposure of camphor leads to cell swelling. So no Cl⁻ trace could be recorded". That again raised the specificity of camphor on TRPV3. Additionally, do we need to wait a long time to record the Cl⁻ trace using fluorescent dye? if yes, why?

In summary, without clearly proving camphor response (both patch recording and Ca²⁺ imaging) is sololy dependent on TRPV3, the conclusion of this manuscript is discounted.

Response: We exposed the cells to camphor for 20 min in the chloride imaging experiments and we observed bleb formation (cell swelling). However, no detectable changes in chloride concentrations were observed up to that time point. This could be caused by small changes in total intracellular chloride concentrations. However, there should be considerable chloride movement that will not affect the total chloride concentrations. This summarizes our basic interpretation. In addition, the experimental conditions in the chloride imaging and patch-clamp experiments were very different, as the intracellular chloride concentrations were kept high because of the pipette solution in patch-clamp. Therefore, we cannot compare the results of the two experiments side by side.

Reviewer #3 (Remarks to the Author):

Most of my concerns have been addressed in this revision except one regarding a transfection of hTRPV3 C169S mutant that is not responsive to agonist camphor and serves as a control. Instead, authors presented the data in which camphor induced currents were inhibited by blocker dyclonine, which is acceptable.

Reviewer #4 (Remarks to the Author):

General Comments

Although the authors have satisfactorily addressed many but not all concerns. Actually, a new one surfaced based on new data provided to other reviewers. Major issues concern the pharmacological approach and the use of very high concentrations of agents to activate or block TRPV3, or block ANO1. The authors also failed to convincingly discard the possibility that TRPV4 might also be a participant in the responses to Camphor.

Specific Comment

1. I appreciate the efforts undertaken by the authors to perform silencing RNA experiments to attempt to knock down TRPV3, TRPV4 and ANO1. It is surprising that not a single RNA molecule tested failed to knock down any of these ion channels at the protein level but this occurs sometimes, which can be frustrating. The authors elected instead to test the effects of the TRPV3 inhibitor dyclonine. However, they used massive concentrations (50-500 μM) to demonstrate an effect. This agent was shown to block TRPV3 currents with an IC_{50} of only 3.2 μM (Liu Q, Wang J, Wei X, Hu J, Ping C, Gao Y, Xie C, Wang P, Cao P, Cao Z, Yu Y, Li D, and Yao J. Therapeutic inhibition of keratinocyte TRPV3 sensory channel by local anesthetic dyclonine. *eLife* 10: e68128, 2021). This raises then raises the question about specificity. Does 500 μM have any direct effect on either TRPV4 or ANO1 currents? The diclonine experiments do not prove that TRPV4 is not involved in the responses. I had suggested to use the specific and potent TRPV4 inhibitor HC067047 (1-10 μM), which would be a satisfactory alternative to siRNA experiments. The authors must perform these experiments to unequivocally discard the possibility that the activity of Camphor is associated with activation of TRPV4.

Response: We thank the reviewer for pointing this out, and we understand his concern about the nonspecific effects of camphor. When we asked whether TRPV4 contributed to the camphor-induced currents, we found that 10 mM camphor did not activate hTRPV4 and that HC067047, a selective inhibitor of TRPV4, did not suppress the 10 mM camphor-induced current in NHEK. In addition, HC067047 did not affect cell migration in the wound healing assay with NHEK. We have added these data as Suppl. Figs. 3 and 7. These results ruled out the possibility that the currents induced by 10 mM camphor and the results in the wound healing assay in NHEK were caused by activation of TRPV4.

We surprisingly found that dyclonine suppressed the ANO1-mediated currents at 500 μM , but not 100 μM (as shown below).

Experimental conditions:

hANO1 expressing HEK cells

HP: 0 mV (-100 mV to + 100 mV)

Bath solution: NMDG-Cl

Pipette solution: NMDG-Cl (500 nM free Ca^{2+})

The previous Supplementary Figure 3 in which 500 μM dyclonine almost completely inhibited 10 mM camphor-induced currents can be interpreted as a mixture of the inhibition of TRPV3-mediated and ANO1-mediated currents. Therefore, we incubated NHEK cells with 100 μM dyclonine for 5 minutes and no current activation was observed even in the presence of 10 mM camphor, suggesting that TRPV3 does contribute to the camphor-induced currents.

Accordingly, we have replaced the previous Suppl. Fig. 3 with a new one (re-revised Suppl. Fig. 4).

2. In the rebuttal, the authors provided a new figure showing that both T16A and Ani9 failed to alter Ca^{2+} responses elicited by 6 mM Camphor. First and foremost, if these two channels are so close to each other, wouldn't you expect a very large effect considering the large Ca^{2+} responses seen? Second, intracellular Cl^- potentially could change (or not), but the more important relationship between the two is related to the effects of ANO1 on membrane potential. Because E_{Cl} is very negative and presumably RMP as well, then blocking ANO1 should depolarize the cells leading to less Ca^{2+} entry through TRPV3 due to a decreased driving force for Ca^{2+} . Third, since 6 mM Camphor produced a large Ca^{2+} response, why wasn't ANO1 activated? In your response to reviewer #3, 3 or 6 mM Camphor had no effect on the current. This makes little pharmacological sense. The difference between 6 and 10 mM on a log scale is very small. Is the higher concentration producing some deleterious non-specific

effects (the authors mentioned swelling of the cells), which are then partially blocked by the high concentrations of agents used?

Response: Last time we attached the data taken from HEK293T cells, which do not express ANO1. We apologize for the mistake. We now have attached new data with 30 μ M Ani9 in NHEK in the presence of 10 mM camphor, but not 6 mM camphor, to address the reviewer's point as shown below.

However, 30 μ M Ani9 did not change the intracellular Ca^{2+} concentrations caused by 10 mM camphor. In terms of the first question, we do not expect large changes in intracellular Ca^{2+} concentrations due to ANO1. Although the reviewer expects changes in intracellular chloride concentrations upon activation of ANO1, it was not the case, as supported by the chloride measurement experiment and stated in the response to the concerns of Reviewer #1. Therefore, such small changes in the intracellular chloride concentrations would not change the membrane potentials and driving force for the TRPV3 channels.

We do not have 6 mM camphor data in the Ca^{2+} -measurements conducted with NHEK. Accordingly, we have no idea regarding the difference in dose-dependency between patch-clamp and Ca^{2+} -imaging experiments. The increase in intracellular Ca^{2+} shown above should cause TRPV3 and following ANO1 current activation.

3. In their answer to my 2nd major point about the biophysical properties of ANO1 when triggered by activation of TRPV3 with Camphor, the authors performed a simple Ca^{2+} clamp (500 nM) experiment to show that the currents were time-dependent. I never questioned the functional expression of ANO1 in these cells. All I asked for was for the authors to demonstrate that activation of TRPV3 with Camphor leads to the appearance of time-dependent currents consistent with ANO1. The authors should use a repetitive step protocol, rather than a ramp protocol, to show that time-dependent CaCC currents are activated during exposure to Camphor.

Response: We misunderstood the meaning of the reviewer's original point.

We did an experiment with NHEK using 10 mM camphor as the reviewer pointed out, which should cause TRPV3 activation, followed by ANO1 activation. We used 100 nM intracellular Ca^{2+} in this experiment. The Camphor-induced currents displayed slow activation kinetics at positive potentials with outwardly rectifying properties and slow recovery after step-pulses, data consistent with the ANO1 channels.

We have added this new data in Suppl. Fig. 5.

4. Figure 3: The authors used 30 μM Ani9 to block the current induced by Camphor. This concentration is excessively high. Ani9 was shown to block ANO1 currents with an IC_{50} of only ~ 110 nM (Seo Y, Lee HK, Park J, Jeon DK, Jo S, Jo M, and Namkung W. Ani9, A Novel Potent Small-Molecule ANO1 Inhibitor with Negligible Effect on ANO2. PLoS One 11: e0155771, 2016) and investigators in the field generally use 1 μM , which has been shown to abolish currents in most cell types. This questions whether the current activated by Camphor is entirely mediated by TRPV3 and ANO1. Since Camphor elicits swelling, is the conductance also comprising a swelling-activated Cl^- current? Does tamoxifen block the current induced by Camphor?

Response: As the reviewer pointed out, the Ani9 concentration used here looks a little bit high. Tamoxifen suggested by the reviewer cannot be used because tamoxifen has been reported to suppress ANO1 (PMID: 18724360). We have never seen the volume increase in the whole-cell patch-clamp experiments while we observed bleb formation, a kind of cell volume increase, in the chloride concentration measurement. Therefore, we are not concerned about the involvement of swelling-activated Cl^- currents in the patch-clamp experiments. In terms of the essential question whether the current activated by camphor is entirely mediated by TRPV3 and ANO1, we believe that other data strongly support it.

Reviewers' comments:

Reviewer #1 (Remarks to the Author):

The author did some experiments but still can't solve the concerns raised previously. Dyclonine is not a selective TRPV3 inhibitor. There is a better TRPV3 inhibitor just published in British Journal of Pharmacology (PMID: 35771623). you should be able to buy that. But again, over expression of GOF mutation or silencing TRPV3 (or use TRPV3 KO) keratinocyte may provide solid evidence to address the conclusion.

Reviewer #4 (Remarks to the Author):

Thank you for your adequate response.

Reviewer #1

The author did some experiments but still can't solve the concerns raised previously. Dyclonine is not a selective TRPV3 inhibitor. There is a better TRPV3 inhibitor just published in British Journal of Pharmacology (PMID: 35771623). you should be able to buy that. But again, over expression of GOF mutation or silencing TRPV3 (or use TRPV3 KO) keratinocyte may provide solid evidence to address the conclusion.

Response: We thank the reviewer for suggesting to use a better TRPV3 inhibitor, scutellarein (Br. J. Pharmacol. 179 (20): 4792-4808, 2022) which was reported to selectively inhibit TRPV3. Unfortunately, however, scutellarein was found to also inhibit ANO1 as shown below, suggesting that the compound cannot be used to prove the TRPV3-ANO1 interaction.

HEK 293 cell expressing human ANO1

Ramp-pulse: -100 mV to +100 mV (HP: 0 mV)

Bath solution: 140 mM NMDG-Cl

Pipette solution: 140 mM NMDG-Cl (500 nM free Ca²⁺)

Then, we decided to use the skin keratinocytes isolated from mouse tail. We observed camphor-induced large chloride currents in WT keratinocytes while such currents were never observed TRPV3-deficient keratinocytes in the NMDG/chloride pipette and bath solutions. In addition, the camphor-induced currents were inhibited by Ani9. These data strongly support the TRPV3-ANO1 interaction. Accordingly, we have included these data

as a new Figure 4.

Although the reviewer suggested the experiments with GOF TRPV3 mutants, we don't think that such experiments provide the useful information to prove the TRPV3-ANO1 interaction.

REVIEWERS' COMMENTS:

Reviewer #1 (Remarks to the Author):

I guess authors have answered my concerns. There are many typos need to be corrected. It can be accepted after correction.